# Adipose stem cells are sexually dimorphic cells with dual roles as preadipocytes and resident fibroblasts

Martin Uhrbom [1,2] ✉, Lars Muhl [1,3], Guillem Genové[1], Jianping Liu [1], Henrik Palmgren[4], Ida Alexandersson [2], Fredrik Karlsson[5], Alex-Xianghua Zhou [4], Sandra Lunnerdal[2], Sonja Gustafsson[1], Byambajav Buyandelger[1], Kasparas Petkevicius[2], Ingela Ahlstedt[2], Daniel Karlsson[2], Leif Aasehaug[6], Liqun He [7], Marie Jeansson [1], Christer Betsholtz [1,7,8] ✉ & Xiao-Rong Peng [2,8] ✉

Cell identities are defined by intrinsic transcriptional networks and spatio-temporal environmental factors. Here, we explored multiple factors that contribute to the identity of adipose stem cells, including anatomic location, microvascular neighborhood, and sex. Our data suggest that adipose stem cells serve a dual role as adipocyte precursors and fibroblast-like cells that shape the adipose tissue's extracellular matrix in an organotypic manner. We further find that adipose stem cells display sexual dimorphism regarding genes involved in estrogen signaling, homeobox transcription factor expression and the renin-angiotensin-aldosterone system. These differences could be attributed to sex hormone effects, developmental origin, or both. Finally, our data demonstrate that adipose stem cells are distinct from mural cells, and that the state of commitment to adipogenic differentiation is linked to their anatomic position in the microvascular niche. Our work supports the importance of sex and microvascular function in adipose tissue physiology.

Adipose tissues (AT) comprise white (W) and brown (B) AT and putative intermediates that play critical roles in systemic metabolism through regulation of energy utilization, adaptive thermogenesis, and adipokine release[1–3]. Maladaptive expansion of WAT from overnutrition poses a significant risk for type-2-diabetes (T2D), cardiovascular disease (CVD), and overall mortality[4,5]. Efforts to deepen the understanding of mechanisms that regulate cellular identity, heterogeneity, and developmental fate of AT resident cells can have

profound implications for the identification of future therapeutic interventions for the treatment of obesity and T2D[6–8].

To accommodate the need for variable nutrient storage and energy mobilization, WAT is one of the most dynamic tissues in the adult mammal. Expansion of WAT involves both cellular hypertrophy (increased adipocyte size) and hyperplasia (increased adipocyte number), the latter resulting from differentiation of resident adipose tissue progenitor cells[9,10]. Region-specific expansion of WAT displays

[1]Department of Medicine, Huddinge, Karolinska Institutet Campus Flemingsberg, Neo building, 141 52 Huddinge, Sweden. [2]Bioscience Metabolism, Research and Early Development Cardiovascular, Renal and Metabolism, BioPharmaceuticals R&D, AstraZeneca, Gothenburg, Sweden. [3]Centre for Cancer Biomarkers CCBIO, Department of Clinical Medicine, University of Bergen, 5020 Bergen, Norway. [4]Bioscience Renal, Research and Early Development Cardiovascular, Renal and Metabolism, BioPharmaceuticals R&D, AstraZeneca, Gothenburg, Sweden. [5]Data Sciences & Quantitative Biology, Discovery Sciences, R&D AstraZeneca, Gothenburg, Sweden. [6]Bioscience Cardiovascular, Research and Early Development Cardiovascular, Renal and Metabolism, BioPharmaceuticals R&D, AstraZeneca, Gothenburg, Sweden. [7]Department of Immunology, Genetics and Pathology, Uppsala University, 751 23 Uppsala, Sweden. [8]These authors contributed equally: Christer Betsholtz, Xiao-Rong Peng. ✉e-mail: martin.uhrbom@ki.se; christer.betsholtz@ki.se; Xiao-Rong.Peng@astrazeneca.com

strong sexual dimorphism in most mammals and correlates with differences in energy metabolism and disease risks. Women in the premenopausal age tend to store fat predominantly in subcutaneous (sc) WAT, which confers protective effects against obesity-related metabolic dysfunction. Conversely, men are prone to expand visceral (v) WAT depots, which is associated with an increased risk of T2D and CVD[11–16]. The underlying molecular mechanisms driving these sex differences remain largely unknown, although homeostatic control by sex hormones and developmental imprinting of cell-intrinsic properties have been implicated[10].

More than 50% of the cells in AT are stromal, including endothelial cells, vascular mural cells (a unifying term for pericytes and smooth muscle cells), fibroblasts, and resident immune cells[17]. Recent advances in technologies such as fluorescence-activated cell sorting (FACS) and single-cell RNA sequencing (scRNA-seq) have provided new insights into the different WAT cell types suggesting that mechanisms governing WAT expansion are more complex than previously anticipated and involve different populations of adipose stem cells (ASC). Two or three subpopulations of ASC have been identified in vWAT and scWAT in mice and scWAT in humans albeit with some differences in claims regarding functional properties and adipogenic potential[6,18–21].

Here, we used scRNA-seq to transcriptionally profile the stromal vascular fraction (SVF) of perigonadal (pg)WAT, a type of vWAT, from male and female *Pdgfrb*-GFP transgenic reporter mice. We find that pgWAT ASC resemble fibroblasts present in skeletal muscle and heart and can be separated into three ASC subtypes consistent with previously proposed ASC classification[6]. We also find that pgWAT ASC exhibits distinct sex-specific gene expression signatures relevant to *Hox* gene expression and vaso-regulatory functions. Finally, we distinguish blood vessel-associated ASC from mural cells and show different ASC subtype features and sex-specific adipogenic differentiation propensity ex vivo.

By integrating multiple intrinsic and micro-environmental variables defining ASC identities, our findings shed light on WAT sexual dimorphism and spatial relationships between ASC and vascular cells in the WAT niche.

## Results

### Cell classes in the stromal vascular fraction of perigonadal white adipose tissue

Stromal vascular fraction cells were collected from pgWAT of 12 to 20-week-old female and male transgenic *Pdgfrb*$^{GFP}$ reporter mice using fluorescent-activated cell sorting (FACS) or CD31 and DPP4 antibody panning (Fig. 1a). ScRNA-seq was performed on a total of 3,261 cells using the SmartSeq2 (SS2) protocol[22]. Clustering of single-cell transcriptomes using the Seurat package[23] resulted in 17 cell clusters (Fig. 1b). Single-cell transcriptome clustering was visualized using UMAP (uniform manifold approximation and projection) plots (Fig. 1b and Supplementary Table 1) and hierarchical clustering based on the Pearson's correlation coefficient calculated from the scaled average expression of the marker genes for each cluster (hereon referred to as Pearson's r) (Fig. 1c). The two methods indicated similar relatedness between the clusters.

To provide provisional annotations to the 17 clusters, we compared cluster-enriched transcripts with known cell type-specific markers[24–27](Fig. 1c). This suggested that clusters #0, 2, 3, 4, 5, 6, 7, 10, and 13 contained fibroblasts-like cells positive for e.g. *Pdgfra, Col1a1, Dcn,* and *Lum*. Cluster #12 contained pericytes positive for e.g. *Kcnj8, Abcc9, Rgs5,* and *Higd1b*. Cluster #14 contained vascular smooth muscle cells (VSMC) positive for e.g. *Acta2, Tagln, Myh11,* and *Mylk*. Clusters #1, 8 and 9 contained blood vascular endothelial cells (EC) positive for e.g. *Pecam1, Cdh5, Kdr,* and *Cldn5*. Cluster #15 contained lymphatic EC positive for e.g. *Prox1, Flt4, Lyve1,* and *Ccl21d*. Cluster #16 contained epithelial cells positive for e.g. *Epcam, Krt18,* and *Fgfr4*. Cluster #11 contained

macrophages positive for e.g. *Cd68, Cd14, Ccr5,* and *Cd163*. Cluster 9 and 13 displayed high levels of mitochondrial gene enrichment which indicates damaged/stressed cells, the two clusters were therefore removed from further analysis (Supplementary Fig. 1a). Because fibroblasts and pericytes are closely related and display few unique markers, we applied a previously assigned 90-gene signature containing 45 fibroblast-enriched and 45 mural cell-enriched genes[25] to support our provisional annotations (Fig. 1c).

Fibroblast-like cells were the most abundant SVF cell type in our dataset. Fibroblasts in other organs, including heart, skeletal muscle, colon, bladder, and lung show extensive organotypic gene expression[25]. A comparison of the pgWAT fibroblast-like cells (clusters #0, 2, 3, 4, 5, 6, 7, 10, and 13) to skeletal muscle and heart fibroblasts[25] revealed separation according to organ-of-origin using both UMAP and Pearson's r plots based on the 1000 most variable genes (Supplementary Fig. 1b and Supplementary table 2), although all cells shared the 45-gene fibroblast signature (Fig. 1c). We next investigated if this organotypicity reflected differential expression of any particular class of genes. Genes for the *matrisome*, which includes extracellular matrix (ECM) and ECM-modulating proteins[28], caused a similar dispersal of fibroblast clusters as the 1000 most variable transcripts. In contrast, genes encoding other functional categories of proteins caused markedly less dispersal (Supplementary Fig. 1b). This suggests that the organotypicity of pgWAT fibroblast-like cells mainly reflects differential expression of matrisome genes in agreement with previous conclusions regarding the transcriptional basis for fibroblast differences between other organs (Fig. 1d)[25].

In addition to matrisome differences, pgWAT fibroblast-like cells distinguished from heart and muscle fibroblasts by expressing key genes in adipogenesis and lipid metabolism such as *Pparg, Fabp4, Plin2* and *Adipoq* (Supplementary Fig. 1c). Ingenuity Pathway Analysis (IPA) indeed suggested that pgWAT fibroblast-like cells display enrichment for molecular functions associated with lipid metabolism (Fig. 1e). This was confirmed in previously published WAT scRNA-seq datasets (Supplementary Fig. 1d and Supplementary Table 3)[18–20,29–32]. We next asked if the pgWAT fibroblast-like cells correspond to ASC (a.k.a. pre-adipocytes). Three subtypes of ASC have previously been described, called ASC1a, ASC1b and ASC2[6]. Using signature markers, we found that all fibroblast-like clusters in our data matched the gene expression signatures of either ASC1a, ASC1b, or ASC2 (Fig. 1f).

While these data provide evidence that ASC correspond to fibroblast-like cells, also murals cells have been proposed to be pre-adipocytes[33–35]. When comparing pgWAT fibroblast-like and mural cells we found stem cell marker *Itgb1* (a.k.a *Cd29*) was expressed by both cell-types, whereas *Cd34/Ly6a* were expressed by fibroblasts and *Mcam* by mural cells (Fig. 1g). Most markers of ASC (Fig. 1f), lipid metabolism and adipogenesis (Supplementary Fig. 2a,b) were enriched in the fibroblast-like cells. *Pparg, Plin2,* and *Fabp4* were equal or higher in pericytes (Supplementary Fig. 2b) but because these genes had their highest expression in EC, we asked if contamination of pericytes by EC cell fragments, a commonly observed phenomenon[24,36], could explain the presence of *Pparg* in our WAT pericytes. In support of this, we noted the presence of numerous canonical EC markers (*Pecam1, Ptprb, Cdh5, Tie1, and Cldn5*) in pericytes at levels matching their level of *Pparg*, i.e. about 50% of that seen in EC (Supplementary Fig. 2b). Not all pericytes were equally EC-contaminated, and after removal of pericytes positive for *Pecam1, Ptprb, Cdh5, Tie1* and *Cldn5*, the remaining pericytes showed low expression of *Pparg, Plin2* and *Fabp4* (Supplementary Fig. 2a, b). Therefore, the abundance of *Pparg, Plin2,* and *Fabp4* in pgWAT pericytes likely reflects contamination by EC.

Taken together, matrisome and adipogenic gene expression suggests that pgWAT fibroblast-like cells fulfill a dual role to shape

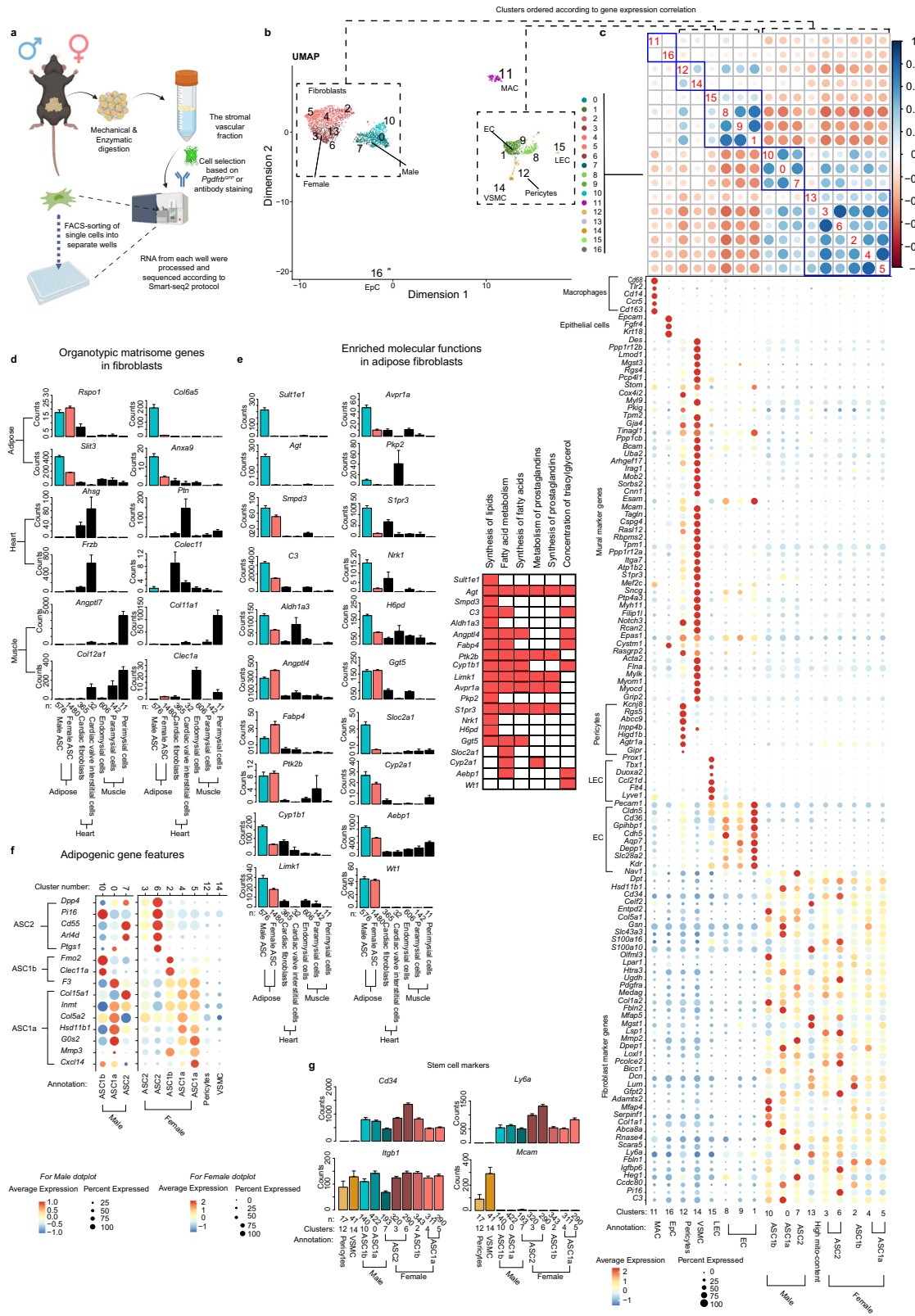

the ECM that provides structural support to WAT (i.e. act as resident tissue fibroblasts) and to act as a reservoir of adipocyte precursors. Whether pgWAT mural cells contribute to adipogenesis as more distant progenitors of pre-adipocytes remains to be addressed.

Previous studies have suggested that ASC2 represents less committed and more multipotent adipocyte progenitors, whereas ASC1a represents a more committed stage of adipocyte differentiation[21]. The position of ASC1b cells in adipocyte differentiation will be discussed below. Because all clusters of pgWAT fibroblast-like cells in our dataset

**Fig. 1 | Outline and cell type characterization. a** Overview of methodology, pgWAT from both female (n = 5) and male (n = 3) mice. Created with BioRender.com released under a Creative Commons Attribution-NonCommercial-NoDerivs 4.0 International license (https://creativecommons.org/licenses/by-nc-nd/4.0/deed.en). **b** Seurat clustering of complete dataset (17 clusters) and UMAP dimensional reduction visualization. **c** Hierarchical ordering of clusters based on Pearson's r value of the scaled average expression of marker gene expression in each cluster with cell-type annotation of clusters and dotplot of selected marker genes. The 45-gene signature for mural and fibroblasts, respectively, as defined by Muhl et al.[25], are displayed in the dotplot. **d** Organotypic matrisome genes in fibroblasts from pgWAT, heart and skeletal muscle. **e** Enriched molecular functions

in adipose ASC when compared to fibroblasts from heart and skeletal muscle, barplot shows gene expression of the enriched genes involved in each function. Red mark in the spreadsheet means that the gene is represented in the molecular function and white mark means that it is not represented. **f** Dotplot over adipogenic gene features of fibroblasts/ASC versus mural cells in pgWAT. **g** Barplots over common stem cells markers used to identify mesenchymal stem cells. Data are presented as mean values +/- SEM. n represent number of cells for scRNA-seq data. Abbreviations: ASC= Adipose stem cell, EpC = Epithelial cells, EC = Endothelial cells, LEC = Lymphatic EC, MAC = Macrophages, Mito=mitochondrial, pg= perigonadal white adipose tissue, SEM=Standard error of the mean,UMAP=Uniform Manifold Approximation and Projection and VSMC = Vascular smooth muscle cells.

matched previously assigned ASC categories (Fig. 1f), we will in the following refer to them as ASC.

## Marker gene signature of sexually dimorphic ASC

For each ASC category (ASC1a, ASC1b, ASC2), we found at least two clusters located in different UMAP islands reflecting male or female origin (Figs. 1b, f and 2a). ASC1a cells (enriched with *Col15a1, G0s2* and *Cxcl14*) were found in cluster 0 (male), and 4 and 5 (female). ASC1b cells (enriched with *Clec11a* and *Fmo2*) were found in cluster 10 (male) and 2 (female). ASC2 cells (enriched with *Dpp4, Cd55* and *Arl4d*) were found in clusters 7 (male) and 3 and 6 (female). Some separation in UMAP between males and females was also observed for EC and macrophages but less conspicuously compared to ASC (Fig. 2a). This suggests that the sexual dimorphism of ASC go beyond the sex-specific expression of Y-chromosome genes and X-chromosome inactivation-associated genes present in all cells.

To validate the sexually dimorphic ASC signatures in independent experiments, we performed bulk RNA-seq on ASC subpopulations isolated by FACS as CD45−/CD31−/CD34+/DPP4± cells from adult male and female iWAT and pgWAT. CD45−/CD31−/CD34+ selection enriches for ASC[37], while DPP4± distinguishes ASC2 (DPP4+) from ASC1a/b (DPP4−) (Supplementary Fig. 3a). We also performed bulk RNA-seq on mature adipocytes from the same mice. Overall, the bulk RNA-seq signatures matched those established by scRNA-seq. High sequence counts for fibroblast markers (e.g. *Dcn, Lum*) and low counts for markers of other cell types (e.g. *Cd68, Pecam1, Kcnj8, Cspg4, Prox1, Pecam1, Lep*) supported purity of the isolated ASC, and the enrichment of marker genes for ASC1a, ASC1b and ASC2 matched between scRNA-seq and bulk RNA-seq data (Supplementary Figs. 4a, b).

Using strict criteria for defining differentially expressed genes (DEGs) (see Methods), we assigned a core set of 36 sexually dimorphic DEGs in ASC identified in both scRNA-seq and bulk RNA-seq data (Fig. 2c and Supplementary Table 4), the latter obtained from both pgWAT and iWAT. When limiting the comparison to pgWAT, an additional 104 sexually dimorphic DEGs were identified (Fig. 2c). Restricting the comparison to bulk RNA-seq samples from pgWAT and iWAT, 29 additional sexually dimorphic DEGs were suggested (Fig. 2c). Five of 36 sexually dimorphic DEGs were sex-chromosome encoded (X chromosome: *Xist, Heph, Prrg3* and Y chromosome: *Ddx3y, Eif2s3y*). Only one of the 36 genes was common between ASC and EC (*Xist*, Fig. 2d). We conclude that the sexually dimorphic DEG pattern is largely cell-type specific and includes several genes associated with lipid handling that are enriched in male ASC (but not in EC) including *Sult1e1, Agt, Avpr1a* and *S1pr3* (Figs. 1e and 2c).

To further validate the mouse ASC sexually dimorphic genes using independent data, we explored the publicly available *Tabula Muris* scRNA-seq dataset, comprising cells from 20 different organs[38]. We selected inguinal (i), perigonadal, and mesenteric (m) cells annotated by the authors[38] as mesenchymal stem cells (MSC, a common term for fibroblast-like cells[36]) and found sex differences in the *Tabula Muris* iWAT and pgWAT MSC that matched our pgWAT ASC data across the core set of 36 sexually dimorphic DEGs, including most of the additional 107 DEGs specific to pgWAT as well as the 29 DEGs restricted to

bulk samples (Fig. 2d, Supplementary Figs. 5a−c and Supplementary Table 5). The *Tabula Muris* mesenteric WAT MSC also matched our pgWAT profile, however, with the exceptions such as *Ptpn5, Pgr, Slc25a30, Sult1e1, Agt* and *Heph*. The similarities and differences in sexually dimorphic gene expression between the different WAT depots may be biologically relevant. For example, male enrichment of *Sult1e1*, encoding a sulfotransferase involved in the inactivation of estradiol[39], may inhibit mammary gland formation in male iWAT[40].

To investigate the impact of sex hormones on the expression of the 36-gene core set of sexually dimorphic DEGs, we performed bulk RNA-seq on sorted CD45−/CD31−/CD34+/DPP4± cells from iWAT in castrated/ovariectomized and control mice. The male enriched transcripts *C7, Sult1e1, Agt, Arl4a, Fkbp5, Angpt1, Arhgap24* and *Ace* were reduced in samples from castrated males (Supplementary Fig. 5d), suggesting that their expression is controlled by androgens.

To make a provisional comparison with human, we explored publicly available single-nuclear (sn)RNA-seq data from human WAT[41]. *FKBP5, SVEP1* and *EGFR*, the human orthologs of mouse *Fkbp5, Svep1* and *Egfr* displayed higher expression in human male subcutaneous ASC and mature adipocytes compared to female cells, consistent with the mouse data (Supplementary Fig. 6a). A consistent change in the direction of differential expression between mouse and human was also observed in ASC from human omental (visceral) depot for human orthologs of the mouse sexually dimorphic genes *Fkbp5, Esr1* and *C7* (Fig. 2c and Supplementary Figs. 6a, b). Other orthologs of mouse sexually dimorphic ASC genes were not confirmed (Supplementary Figs. 6a, b), but the significance of this is uncertain owing to the different technical platforms and depth of sequencing data. In conclusion, while confirming part of the mouse sexually dimorphic ASC gene expression pattern, additional and deeper human data will be required for a comprehensive comparison.

One of the most significant DEGs in female mouse pgWAT was *Hoxa10* (Fig. 2b), an observation that prompted a broader analysis of *Hox* transcripts. Male ASC showed enriched expression of *Hox* transcripts with lower numbers (*Hox(abc)1-8*), whereas the opposite pattern was observed in females (enriched expression of *Hox(acd)9-13*) (Fig. 2e). This *Hox* pattern was confirmed in bulk RNA-seq data from pgWAT ASC for 21 out of 25 genes (Supplementary Fig. 6c, d). The *Hox* pattern was not observed in pgWAT EC, but was present in pgWAT MSC from the *Tabula Muris* dataset. Intriguingly, the *Hox* pattern was not observed in iWAT or mesenteric MSC (Fig. 2e and Supplementary Fig. 6c, d). No clear trend towards similar patterns was observed in human subcutaneous and visceral (from the omental depot) ASC (Supplementary Fig. 6e)[41]. The physiological relevance of sexually dimorphic *Hox* gene expression remains to be determined. It may reflect a different developmental history of pgWAT in males and females.

## Sexually dimorphic pathways include RAAS and glucose metabolism disorder

We next used IPA to search for signaling pathways and cellular functions potentially affected by sexually dimorphic gene expression patterns. IPA analysis suggested *Enhanced Renin-Angiotensin-Aldosterone-*

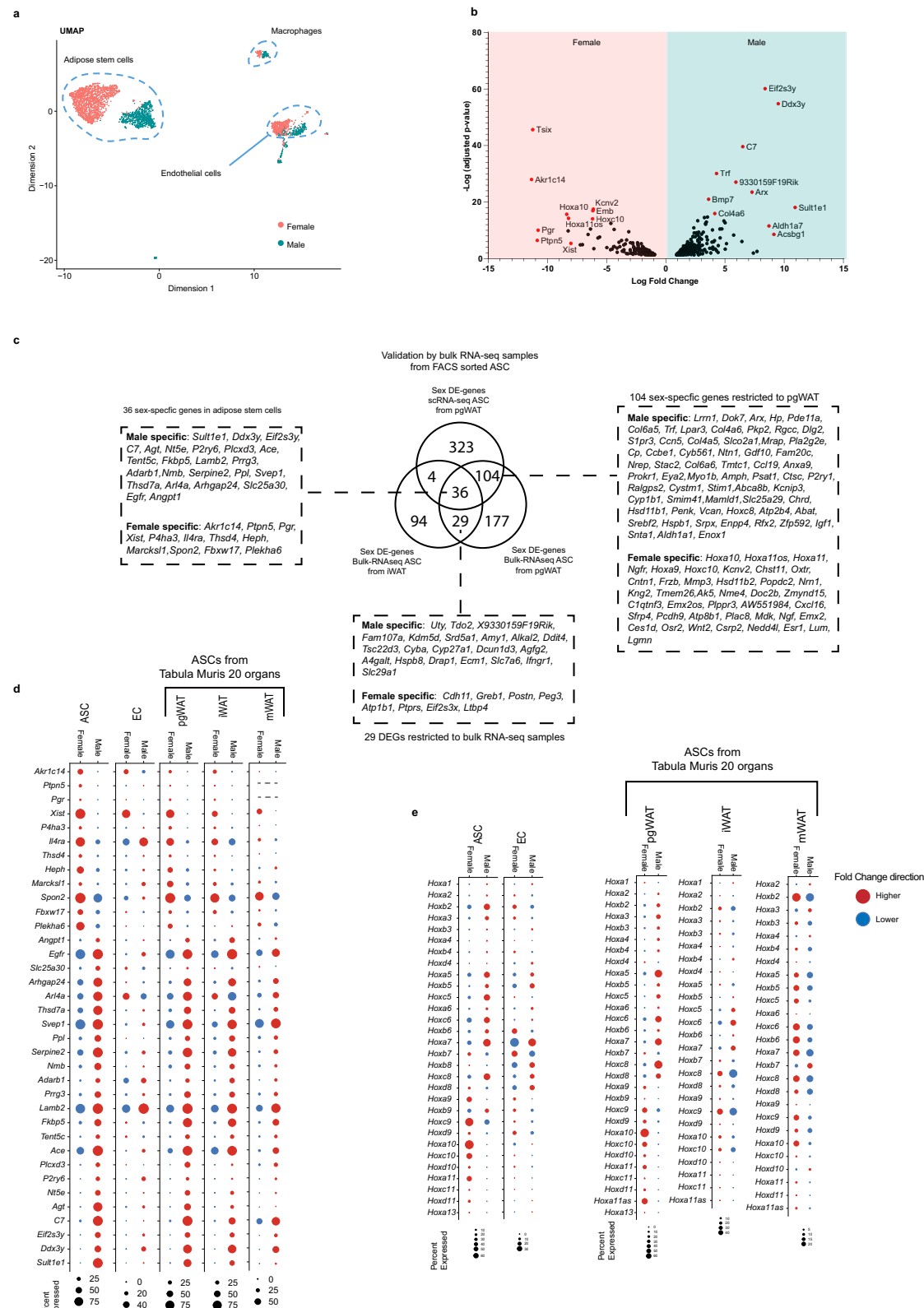

**Fig. 2 | Sexually dimorphic gene signature of adipose stem cells. a** UMAP-projection with male and female cells highlighted. **b** Volcano plot over differentially expressed genes between male and female ASC. Fold changes were calculated by EdgeR-LRT and p-values were adjusted for multipletesting using the Benjamini-Hochberg method. **c** Venn diagram indicating the 36 common differentially expressed genes in scRNA-Seq (from pgWAT) and FACS sorted ASC Bulk RNA-seq samples from both pgWAT and iWAT. The 104 sexually dimorphic DEGs specific to pgWAT and the 29 DEGs specific for bulk RNA-seq samples are also highlighted

**d** Dot plot of the expression of the core set of 36 sexually dimorphic genes in ASC and endothelial cells from our scRNA-seq data with an outlook into the Tabula Muris consortium's scRNA-seq mouse 20 organs database for mesenchymal stem cells in perigonadal, inguinal and mesenteric adipose tissue[38] **e** Same as in **d** but for *Hox* gene expression. Abbreviations: ASC Adipose stem cells, DEGs Differentially expressed genes, i inguinal, LRT likelihood Ratio Test, m mesenteric, pg perigonadal, sc single cell, RNA-seq RNA-sequencing, UMAP Uniform Manifold Approximation and Projection and WAT White adipose tissue.

*System* (RAAS) pathway in male ASC (Fig. 3a). Moreover, one of the top enriched terms for diseases and biological functions associated with the 36-gene core set of sexually dimorphic genes was *Glucose Metabolism Disorder* (Supplementary Figs. 7a, b).

Related to *Renin-Angiotensin-Aldosterone-System*, male ASC showed enriched expression of *Ace*, encoding angiotensin-converting enzyme, and *Agt*, encoding angiotensinogen (AGT). We found that other key genes of the RAAS-system (schematically illustrated in Fig. 3b) were also expressed in WAT but without sexual dimorphism (Fig. 3c). These genes included *Atp6ap2* (encoding the renin receptor) expressed by ASC and adipocytes, *Ctsd* (encoding cathepsin D) expressed across multiple cell types, *Cma1* and *Mcpt4* (encoding chymases) expressed by macrophages, *Enpep* and *Anpep* (encoding aminopeptidases A and -N respectively) expressed by ASC, and *Agtr1a* (encoding angiotensin II receptor) expressed by ASC and mural cells. *Ace2*, encoding angiotensin-converting enzyme 2, which is also the cellular receptor for SARS-CoV-2, was weakly expressed in our RNA-seq data. The role of a putatively increased RAAS signaling in male WAT remains unclear. Angiotensin II (AngII) has been reported to influence adipocyte differentiation and lipolysis[42,43]. However, we found no alteration in basal lipolysis in iWAT explants exposed to 100 ng/ml of AngII (Fig. 3d) or in adipogenic differentiation of SVF cells from the same depot (Fig. 3e) despite AngII receptor expression (Fig. 3f). It is therefore possible that sexually dimorphic RAAS activity plays a role in hemodynamic regulation of WAT rather than having a direct effect on cell differentiation or lipolysis.

Related to enriched terms for diseases and biological functions the term *Glucose Metabolism Disorder* had the highest number of associated genes from the 36-gene core set (18/36) and the low p-value for the term is likely reflecting that the directional change of 17/18 genes in our data was consistent with previous reported data in the literature (Supplementary Figs. 7a–c). Although the IPA software did not assign *Glucose Metabolism Disorder* specifically to male or female, most of the IPA-indicated references report aggravated disorder in males (Supplementary Fig. 7c). For example, one study concluded that knockout of *Fkbp5* (male ASC-enriched) decreases insulin resistance in mice on a high fat diet[44]. Another study found that upregulation of the *Svep1* (male ASC-enriched) was associated with T2D in mice[45].

Genes involved in sex-hormone signaling were among the sexual dimorphic DEGs. Female cells showed enriched expression of progesterone receptor (*Pgr*) and estrogen-receptor alpha (*Esr1*) (Figs. 2b, c and 3g). Conversely, male ASC showed enriched expression of the estrogen inactivator *Sult1e1* (Fig. 2b, c).

## Male ASC1a/b cells have higher adipogenic potential in vitro than their female counterparts

Earlier studies have shown that DPP4 is highly expressed on the cell surface of human preadipocytes, and DPP4 has been suggested to affect both lipid metabolism and cell proliferation[46]. Previous publications have also suggested that DPP4⁺ adipogenic progenitors (ASC2) are less prone to differentiate into mature adipocytes[21]. Our scRNA-seq data included *Dpp4*⁺ (ASC2) and *Dpp4*⁻ (ASC1a/b) cells from both males and females (Fig. 4a). To assess the potential of self-renewal and differentiation of DPP4⁺ and DPP4⁻ cells in SVF preparations from iWAT, we assessed proliferation rate and adipogenic differentiation by exposing confluent cell cultures to insulin alone or a cocktail of adipogenesis-inducing reagents including insulin, dexamethasone, IBMX and pioglitazone. In these experiments, DPP4⁻ (ASC1a/b) cells (Supplementary Fig. 3a) showed low proliferation (Fig. 4b) and high lipid droplet accumulation in the presence of insulin alone (Fig. 4c). Conversely, DPP4⁺ (ASC2) had high proliferation rate (Fig. 4b) and a low lipid droplet accumulation in the presence of insulin alone, which was marginally increased by the full cocktail (Fig. 4c). Marker genes for mature adipocytes (*Lpl, Fabp4, Adipoq, Lep, Pparg*) were all significantly

higher in DPP4⁻ (ASC1a/b) cells than in DPP4⁺ (ASC2) cells (Fig. 4d) after adipogenic differentiation. The higher potential for self-renewal and lower ability to differentiate suggest that ASC2 cells are less committed adipose precursor cells than ASC1a/b.

We next compared the differentiation of ASC isolated from iWAT and pgWAT between sexes using the full cocktail of inducers. Male ASC1a/b from both pgWAT and iWAT showed increased adipogenic differentiation compared to the corresponding cells from females (Fig. 4e, f). No sex difference was observed regarding the (low) propensity of ASC2 to differentiate into adipocytes.

We further asked whether the sex-specific difference in ASC1a/b differentiation was influenced by other SVF cells. To this end, we isolated crude SVF cells from iWAT/pgWAT and applied the same differentiation protocol as for the FACS-sorted ASC. Crude SVF cells contain most of the non-parenchymal cell-types of the depots (ASC, endothelial cells, hematopoietic cells) and are commonly used for studying adipogenesis in vitro. A trend toward higher differentiation was observed in males (Supplementary Figs. 8a, b). Between the two AT depots, iWAT SVF showed higher adipogenic differentiation than pgWAT, which was statistically significant in females (Supplementary Figs. 8a, b).

Because influence of sex on adipogenic differentiation appeared weaker in crude SVF cultures compared to FACS-sorted cells, we asked whether in vitro SVF culturing affected the sex-specific ASC transcriptome. Bulk RNA-seq of confluent crude SVF cultures 4 days after in vitro plating (i.e. at the state of the cells just before differentiation was initiated) showed loss of the 36 sexually dimorphic ASC gene profile and the *Hox* gene expression pattern observed in pgWAT in vivo (Supplementary Figs. 8c, d). Also lost was the sex-specific clustering of transcriptomes seen with the scRNA-seq data (Supplementary Figs. 8e, f). Instead clustering occurred by fat depot origin: iWAT or pgWAT (Supplementary Figs. 8e, f). This was illustrated also at the level of individual genes: iWAT SVF maintained the specific (for iWAT) expression of *Tbx15* whereas pgWAT SVF maintained the specific (for pgWAT) expression of *Tcf21* (Supplementary Fig. 8g, h). High expression of fibroblasts markers (*Col1a1, Col3a1, Dcn, Lum, Pdgfra, Fn1*) and low expression of markers for endothelial cells, macrophages, pericytes and VSMC (Supplementary Fig. 8i) confirmed that ASC are the major cell type of crude SVF cultures. Moreover, markers of ASC2 (*Dpp4* and *Cd55*) were higher in pgWAT SVF than in iWAT SVF in agreement with the lower adipogenic differentiation of pgWAT SVF (Supplementary Fig. 8j). Low expression of *Pparg* in female pgWAT SVF (Supplementary Fig. 8k) may explain the distinct and consistent low differentiation grade of these samples. WNT hormone signaling through frizzled receptors is known to downregulate *Pparg*[47]. We noted that several transcripts of WNT pathway activators were enriched in female pgWAT SVF, some of which were also enriched in FACS-sorted female pgWAT ASC1a/b cells (Supplementary Fig. 9a,b). *Wnt4* was consistently enriched in female pgWAT cells, and *Rspo1*, encoding R-spondin-1 which potentiates WNT signaling, was consistently enriched in pgWAT in both males and females (Supplementary Fig. 9b). *Fzd1*, encoding Frizzled-1 receptor, was expressed in pgWAT, particularly in female ASC1a/b cells, as shown by our scRNA-seq data on FACS sorted cells (Supplementary Fig. 9b).

## Morphological distinction of mural cells and ASC along the adipose microvascular tree

Because pericytes have previously been suggested to constitute ASC, and because we found distinct adipogenic behavior of ASC1a/b and ASC2, we investigated the spatial relationships between ASC and WAT microvessels. We visualized endothelial cells, pericytes, VSMC and ASC in pgWAT isolated from *Pdgfrb*^GFP reporter mice. This mouse strain has previously been used for mural cell imaging[48]. Although *Pdgfrb* is also expressed by fibroblasts, including ASC (Fig. 5a), mural cells typically have stronger *Pdgfrb* expression and display a stronger *Pdgfrb*^GFP

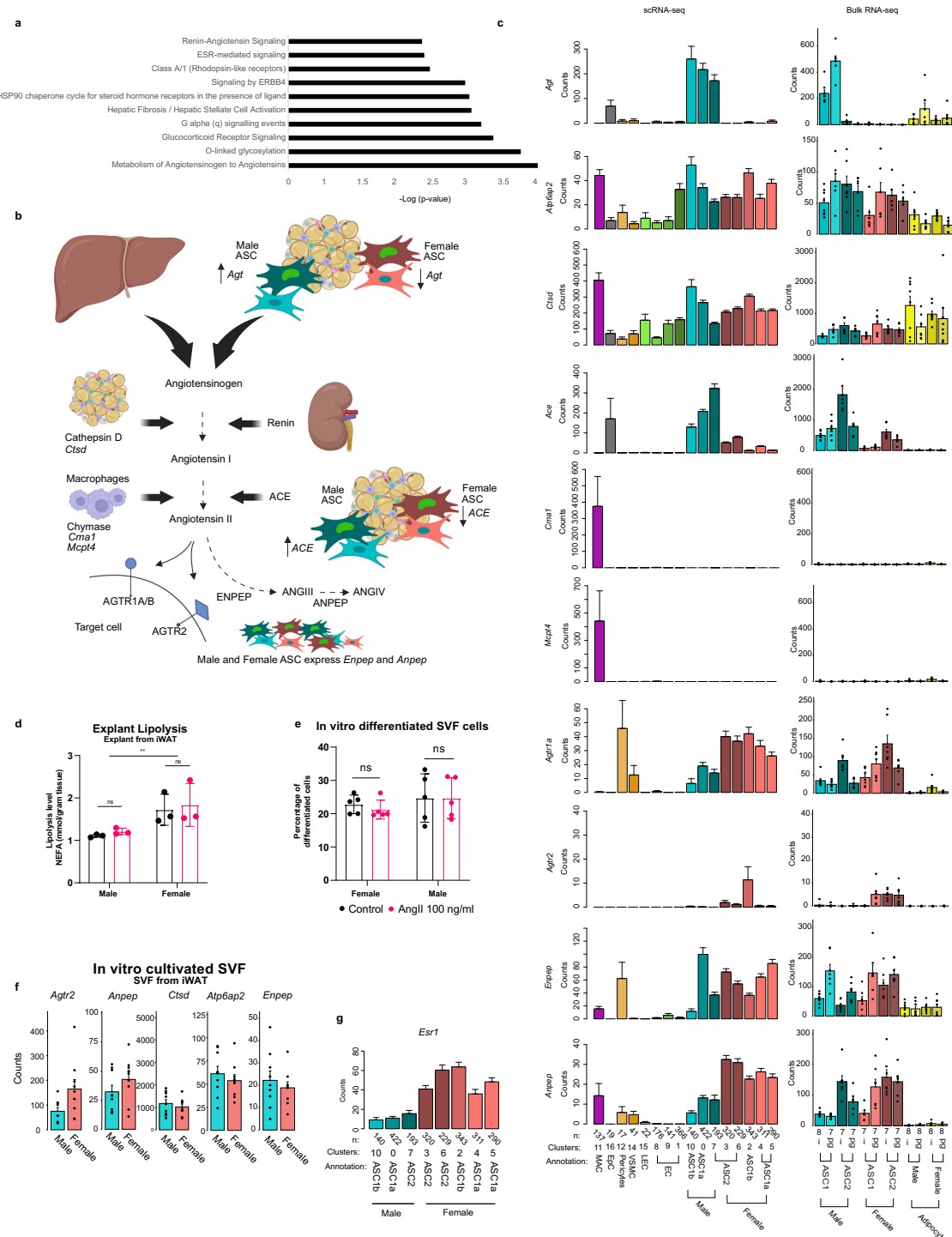

**Fig. 3 | RAAS and glucose metabolism disorder two sexually dimorphic pathways. a** Enriched canonical pathways in male and female ASC based on the core set of 36 sexually dimorphic genes. P-Values are derived from IPA-analysis and based on the right-tailed Fisher's Test. **b** Schematic cartoon over the RAAS-system and the expression of its main components in adipose tissue. Created with BioRender.com released under a Creative Commons Attribution-NonCommercial-NoDerivs 4.0 International license (https://creativecommons.org/licenses/by-nc-nd/4.0/deed.en). **c** Barplots of the expression of RAAS-associated genes in scRNA-seq dataset, FACS sorted ASC populations (bulk RNA-seq) and mature adipocytes (bulk RNA-seq). For all bulk samples *n* = 7 biological replicates except ASC1 male iWAT and mature adipocytes samples (*n* = 8). **d** AngII effect on lipolysis from iWAT explants.

*n* = 3 biological replicates, P-value = 0.0092 **e** AngII effect on in vitro differentiation of crude SVF cells from iWAT. *n* = 5 and represent five independent experiments. **f** Expression of detected RAAS-associated genes in cultivated SVF cells prior to initiation of differentiation. *n* = 9 technical well replicates from three independent experiments **g** Expression of *Esr1* in ASC from pgWAT (scRNA-seq data). Statistics in Fig. 3d, e were calculated with two-way Anova using Sidak's multiple comparisons test. Data are presented as mean values +/- SEM for c, f,g and mean values +/- SD for **d** and **e**. AngII Angiotensin II, ASC Adipose stem cells, ETC electron transport chain, i inguinal, pg perigonadal, RAAS Renin-angiotensin-aldosterone system, sc single cell, SD standard deviation, SEM Standard error of the mean, seq sequencing and WAT White adipose tissue.

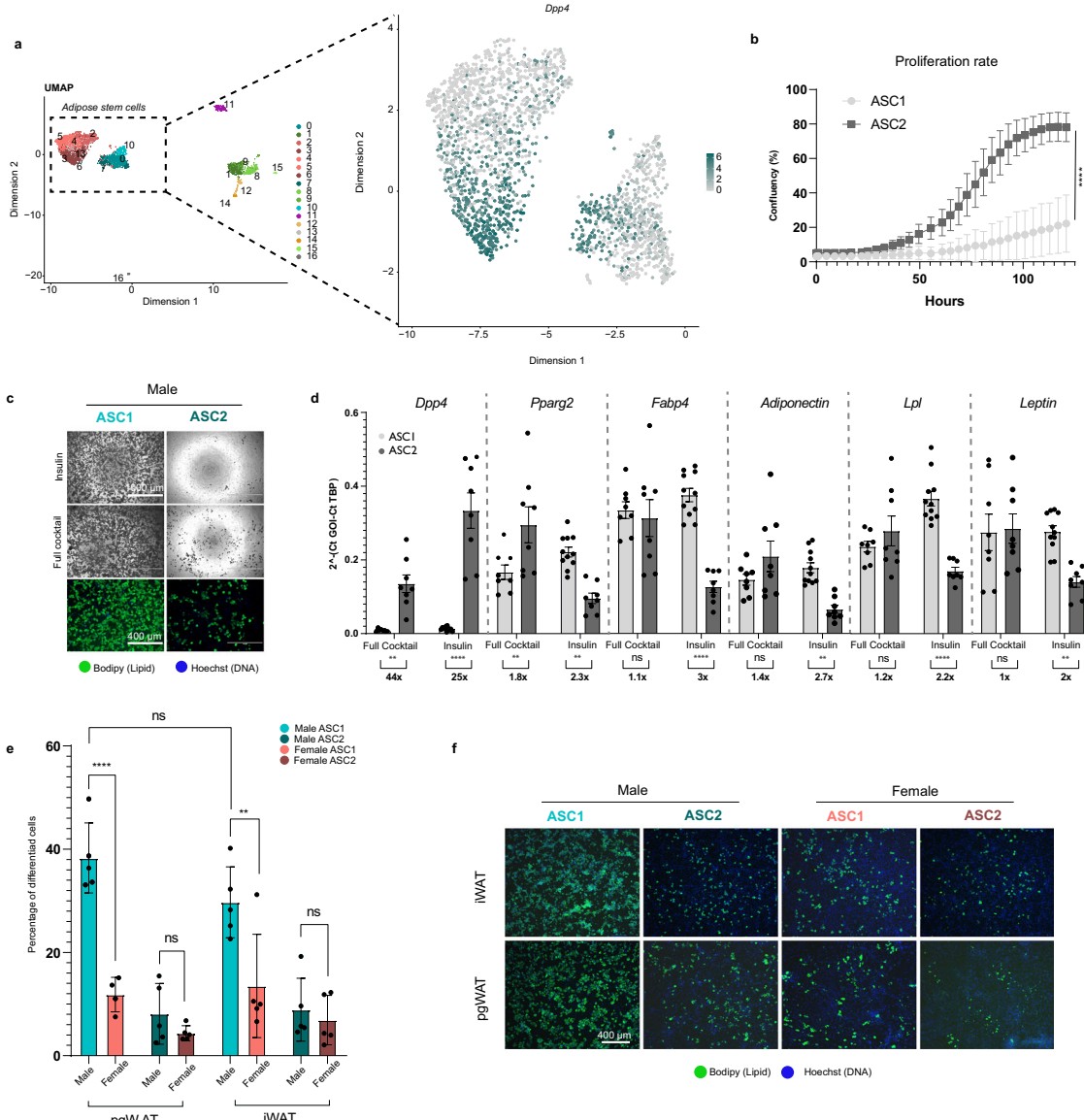

**Fig. 4 | In vitro results. a** *Dpp4* gene expression highlighted in UMAP projection of ASC. **b** Proliferation rate measured in vitro of isolated ASC1 and ASC2 cells. n = 10 for ASC1 and n = 20 for ASC2, n represents technical well replicates, similar results have been obtained in three independent experiments **c** Representative images of In vitro differentiated ASC1 and ASC2 cells using insulin or a full cocktail of adipogenic reagents **d** Expression of *Dpp4* and marker genes for mature adipocytes in in vitro differentiated ASC. *n* = 8 for all groups except ASC1 insulin treated cells for which *n* = 11. n represent technical well replicates from three independent experiments **e** Barplot over the level of differentiation in ASC1 and ASC2 cells from iWAT and pgWAT from adult male and female mice. *n* = 5 biological replicates for all groups except female pgWAT ASC1 for which n = 4. **f** Representative images of

differentiated ASC1 and ASC2 cells from iWAT and pgWAT from adult male and female mice. Statistics in **b** were calculated with a two-sided unpaired t-test for the data points collected at the final time point, t = 12.23, degrees of freedom =28, P-value < 0.001. Statistics in **d** and **e** were calculated with two-way ANOVA and Mixed-effects analysis, respectively, using Tukey's multiple comparison test (Prism). Adjusted P-values for multiple testing were used (*$P$ < 0.0332, **$P$ < 0.0021, ***$P$ < 0.0002 and ****$P$ < 0.0001) for **d** and (**$P$ = 0.0055 and ****$P$ < 0.0001) for **e**. The statistics in **d** were based on the delta Ct-values using *TBP* as a house keeping gene, see source data. Data are presented as mean values +/- SD. ASC Adipose stem cells, i inguinal, pg perigonadal, SD Standard deviation, UMAP Uniform Manifold Approximation and Projection, and WAT White adipose tissue.

signal[25]. In contrast, *Pdgfra* is a broad marker of fibroblasts and typically not expressed by mural cells[25]. We confirmed these expression patterns in our WAT scRNA-seq data (Fig. 5a). We used anti-PDGFRA antibodies to discriminate ASC from mural cells, anti-DPP4 antibodies to visualize the ASC2 population, and anti-CD31 (PECAM-1) to visualize endothelial cells in *Pdgfrb*[GFP] mouse WAT. *Pdgfrb*[GFP+] cells displayed the typical morphologies of pericytes and VSMC adjacent to CD31-labeled endothelium (Fig. 5b). The strong *Pdgfrb*[GFP] signals and long processes adherent to the abluminal side of capillary endothelial cells were consistent with pericytes, as known from other organs. WAT pericytes resembled so-called thin-strand pericytes of the mouse brain[49,50] (Fig. 5b, inset #3). Other *Pdgfrb*[GFP+] cells extended processes

enveloping the vessel circumference; a phenotype consistent with arterial VSMC (Fig. 5b, inset #1). Intermediate mural cell morphologies typical of arteriolar VSMC were also observed (Fig. 5b, inset #2). VSMC with multiple short processes without obvious longitudinal or transversal orientation were observed in venules and veins (Fig. 5b, insets #4-5). Taken together, our observations suggest that AT mural cells display a continuum of morphologies along the arterio-venous axis similar to what has previously been described in brain[24,48].

Anti-PDGFRA antibodies labeled ASC at both perivascular and interstitial locations (Fig. 5c). Of these, the ASC2 subpopulation was identified by anti-DPP4 staining as the, cells with interstitial location (Fig. 5d, arrows). We also noticed a DPP4 positive layer of cells

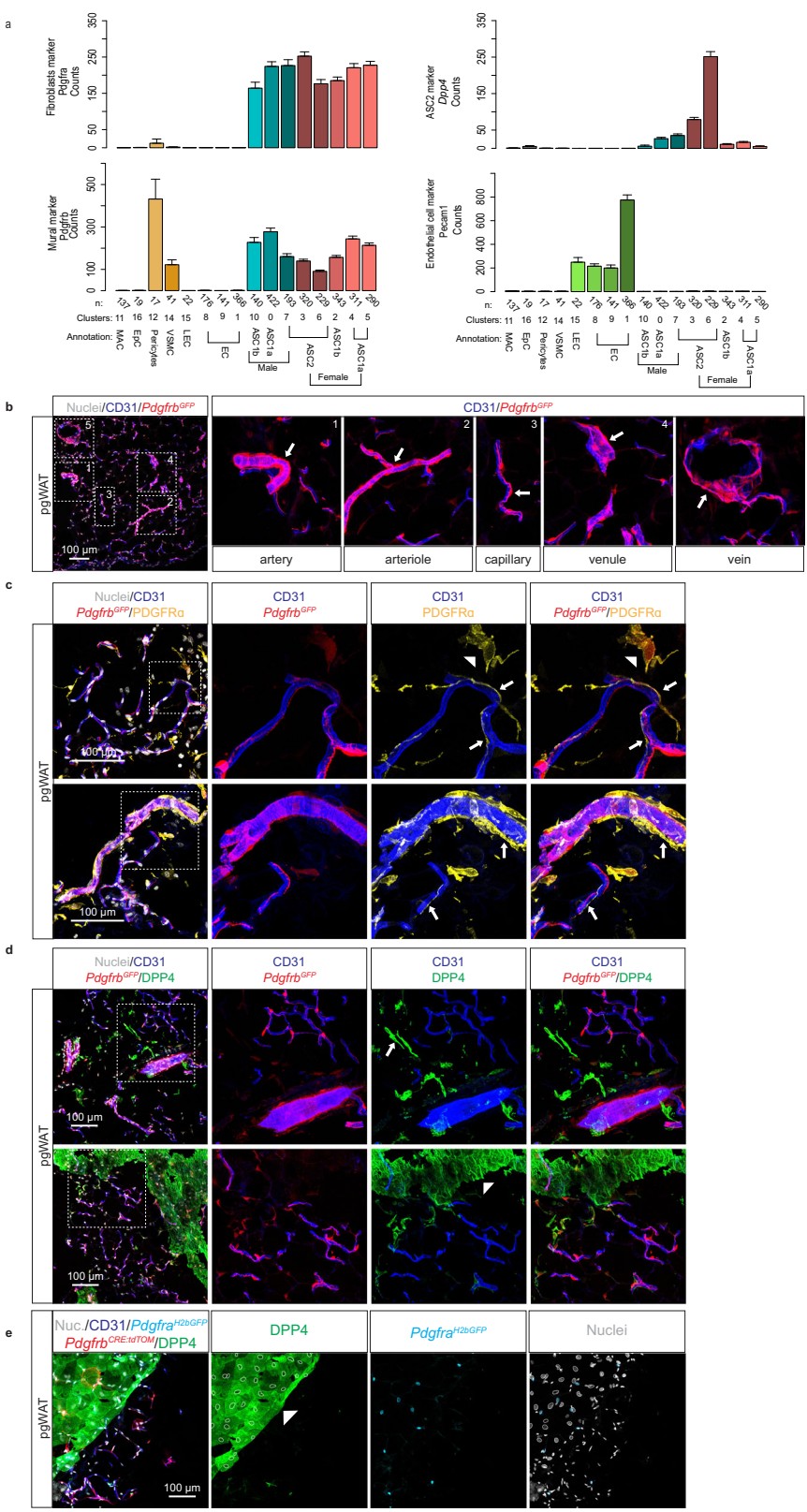

**Fig. 5 | Cell localization in perigonadal white adipose tissue in *Pdgfrb*^GFP reporter mice. a** Barplots of marker gene expression used for cell visualization. Data are presented as mean values +/- SEM. **b** Immunofluorescence staining of pgWAT from *Pdgfrb*^GFP report line for CD31 (also known as PECAM1) displaying mural cells across the arteriovenous axis. **c** Same as in **b** but with staining for CD31 and PDGFRA. Arrows and arrowheads indicate the position of perivascular and interstitial ASC, respectively. **d** Same as in **b** but with staining for CD31 and DPP4. The Arrow and arrowhead indicate the location of interstitial ASC2-population and mesothelial cells, respectively. **e** Maximum intensity projection of slices with mesothelial DPP4 staining in *Pdgfrb*^CRE-tdTOM/*Pdgfra*^H2bGFP reporter mice. Predicted mesothelial nuclei (white circles) were marked based on the DPP4 staining pattern, and *Pdgfra*^H2bGFP-positive nuclei (cyan circles) were marked dependent on the GFP signal. No overlap of mesothelial nuclei and *Pdgfra*^H2bGFP-positive nuclei could be observed. GFP Green fluorescent protein, pgWAT perigonadal White adipose tissue and SEM Standard error of the mean.

covering the surface of the pgWAT depots (Fig. 5d, e, arrowhead). These cells were negative for PDGFRB and PDGFRA had the expected location of mesothelial cells, previously suggested to express DPP4[6]. To further investigate the spatial relationship between ASC subpopulations and vasculature, we performed whole-mount analysis of the pgWAT from *Pdgfrb*[CreERT2:R26tdTomato] /Pdgfra[H2BGFP]reporter mice. We localized all ASC using *Pdgfra*-driven nuclear GFP and, simultaneously, ASC2 by anti-DPP4. The results confirmed the observations on *Pdgfrb*-reporter mice, namely that DPP4- ASC1a/b-cells are located in close vicinity to and partially in direct contact with blood vessels (Fig. 6a–c), whereas most DPP4+ ASC2 cells were located at discernable distance from the vessels (Fig. 6a, b). These spatial relationships were confirmed by 3D image rotation (Fig. 6a, b), or by 3D rendering videos of whole mount preparations. The latter analysis showed that DPP4+ ASC2 cells that appeared blood vessel-associated in 2D were not, judging by 3D rendering (Supplementary Movies 1 and 2). Figure 6c shows a schematic cartoon of the vasculature and ASC subpopulations in pgWAT.

We finally asked if any of the sexually dimorphic ASC genes could be verified at the protein level in ASC at their respective perivascular or interstitial locations. Such analysis is strictly dependent on antibodies that are specific and functional together with other antibodies in immunofluorescence of pgWAT. Here, we found that antibodies against nerve growth factor receptor (NGFR, encoded by the female-specific mRNA *Ngfr*) stained perivascularly located ASC (PDGFRA+) (Fig. 6d) and interstitially located ASC2 (DPP4+) (Fig. 6e) in pgWAT in agreement with the *Ngfr* being one of the 104 DEGs for pgWAT (Fig. 2c).

## Discussion

Establishment of specific ASC populations in different WAT depots likely depends on a combination of factors, including developmental signals, sex and anatomic location. Adipogenesis has been linked to the vascular niche in WAT, a location that harbors several different cell types including mural cells, fibroblasts, and endothelial cells, which have all been suggested as preadipocytes[51–55]. Lineage-tracing is complicated by the shortage of specific pan-fibroblast and pan-mural cell markers. A cross-organ comparison of scRNA-seq data combined with in vitro assays, as presented herein, converge on a fibroblast-like identity for ASC. Based on the expression of canonical fibroblast markers (e.g. *Pdgfra, Col1a1, Dcn*, and *Lum*) and a 90-gene signature for discrimination of fibroblasts from mural cells[25], we conclude that pgWAT ASC are equivalent to WAT fibroblasts. Like fibroblasts in other organs, pgWAT ASC showed organotypic features, as reflected by differential expression of matrisome genes (e.g. *Rspo1, Col6a5, Frzb, Col11a1* and *Col12a1*) alongside genes involved in the regulation of adipogenesis (*Pparg*) and lipid metabolism (*Fabp4, Plin2*). These data suggest that ASC serve a dual role of being adipocyte precursors and tailors of the specific WAT ECM composition. ASC heterogeneity within individual WAT depots has been demonstrated under both basal conditions and after a challenge by obesogenic diet or β3-adrenergic receptor activation[18,56]. Our transcriptional profiles of pgWAT fibroblasts matched previously reported ASC1a, ASC1b, and ASC2 subpopulations[6].

It is increasingly clear that adiposity at different anatomic locations, e.g. subcutaneous, gluteofemoral, and visceral, have distinct metabolic profiles that are strongly influenced by sex, and that these profiles may be more reliable proxies of T2D and cardiovascular disease risks than BMI[11,57,58]. Despite this, hitherto published scRNA-seq studies of mouse WAT either focused on males or lacked specific analysis of sex differences when both sexes were present[6,18–21]. Here, we uncover sexual dimorphism of ASC with putative importance for the metabolic profile of WAT. The sexual dimorphism is observed in ASC transcriptomic signature, as well as in some ex vivo adipogenic behaviors of isolated ASC.

WAT is an important source of AGT and expresses the machinery necessary to generate the vasoconstrictor AngII. Our scRNA-seq data revealed that *Agt* and *Ace* are highly expressed in male ASC. If and how adipose RAAS contributes to obesity-associated hypertension systematically is unclear[59,60]. Given the proximity of ASC to blood vessels, locally generated AngII may regulate microvascular tone. AGTR1, the receptor for AngII, is expressed by pericytes in WAT and other tissues[61]. Our results that high levels of AngII lacked an effect on basal lipolysis and adipogenic differentiation contradicts previous work showing that AngII inhibits lipolysis and impacts differentiation[42,43]. Further work is needed to understand the relative contribution of locally produced AngII and its relationship to AT capillary function or capillary function in peripheral tissue in general, as well as the relevance to human AT biology. The finding of AGTR1, the AngII receptor expression in pericytes is intriguing, since WAT blood flow is regulated between meals[62], and pericytes have been suggested to regulate blood flow in the brain and heart. Hence, a similar role for pericytes may be speculated for adipose tissue[63].

Sexual dimorphic expression of estrogen receptor alpha (*Esr1*) and the estrogen-inactivating enzyme *Sult1e1*, indicates that estrogen-receptor signaling is a strong driver of the sex differences in ASC transcriptomes. Studies of male scWAT adipose progenitor cell transplantation into females suggested that the transplanted cells adopted the behavior of the host during high fat diet, suggesting environmental control in which sex hormones likely play a role[10]. In this context, it was interesting to note that most of the sex-specific transcriptomic differences disappeared when WAT SVF cells were grown in vitro. In marked contrast, some of the conspicuous depot-dependent differences in transcription factor gene expression (*Tbx15* and *Tcf21*) remained in vitro.

Our data suggest that *Hox* genes with numbers below 9 are male pgWAT ASC specific, whereas numbers 9 and above are female specific. Sex -and depot-dependent differences in *Hox* genes expression of AT has been reported previously[64–71]. One of these reports focused on the developmental signature of human abdominal and gluteal subcutaneous adipose tissues in men and women[69]. They found that *Hox* genes with lower numbers (as in our male-derived ASC) were enriched in abdominal depots of both sexes, whereas *Hox* genes with higher number (as in our female-derived ASC) were upregulated in gluteal depot of both sexes. This might reflect different embryonic origins of the ASC in males and females, with or without physiological impact in adults. Either way, the differential expression of *Hox* genes suggests that sexually dimorphic properties of ASC may to some extent be imprinted already during embryonic development and maintained through life.

We further observed that FACS-sorted ASC1 from male inguinal (subcutaneous, iWAT) and perigonadal (visceral, pgWAT) AT depots had higher adipogenic potential in vitro than female counterparts. A similar trend, albeit not statistically significant, was seen in the adipogenic potential of crude SVF from both iWAT and pgWAT. Several factors might contribute to this difference. ASC2 might become dominant over ASC1 in culture due higher proliferation rate. Alternatively, the presence of other cell types may regulate the adipogenic potential of ASC in culture. Transcriptomic analysis of cultured SVF just prior to initiation of differentiation indicated a higher proportion of ASC2-cells and lower expression of *Pparg* in the pgWAT samples. More active WNT-signaling in female pgWAT (higher levels of e.g. *Fzd1, Rspo1* and *Wnt4*) may also underlie this difference. WNT hormones are known inhibitors of adipogenesis that downregulate *Pparg* through canonical WNT signaling[47]. Merrick et al.[21] found a higher proliferation rate and lower adipogenesis in ASC2 compared to ASC1a/b and that ASC2 represents a multi-potent and less committed precursor population that contributes to basal adipogenesis in both visceral and subcutaneous fat depots[56]. Our results concur with this conclusion. Furthermore, Merrick et al. used Lin−/CD142+, Lin−/CD142−/DPP4+ and

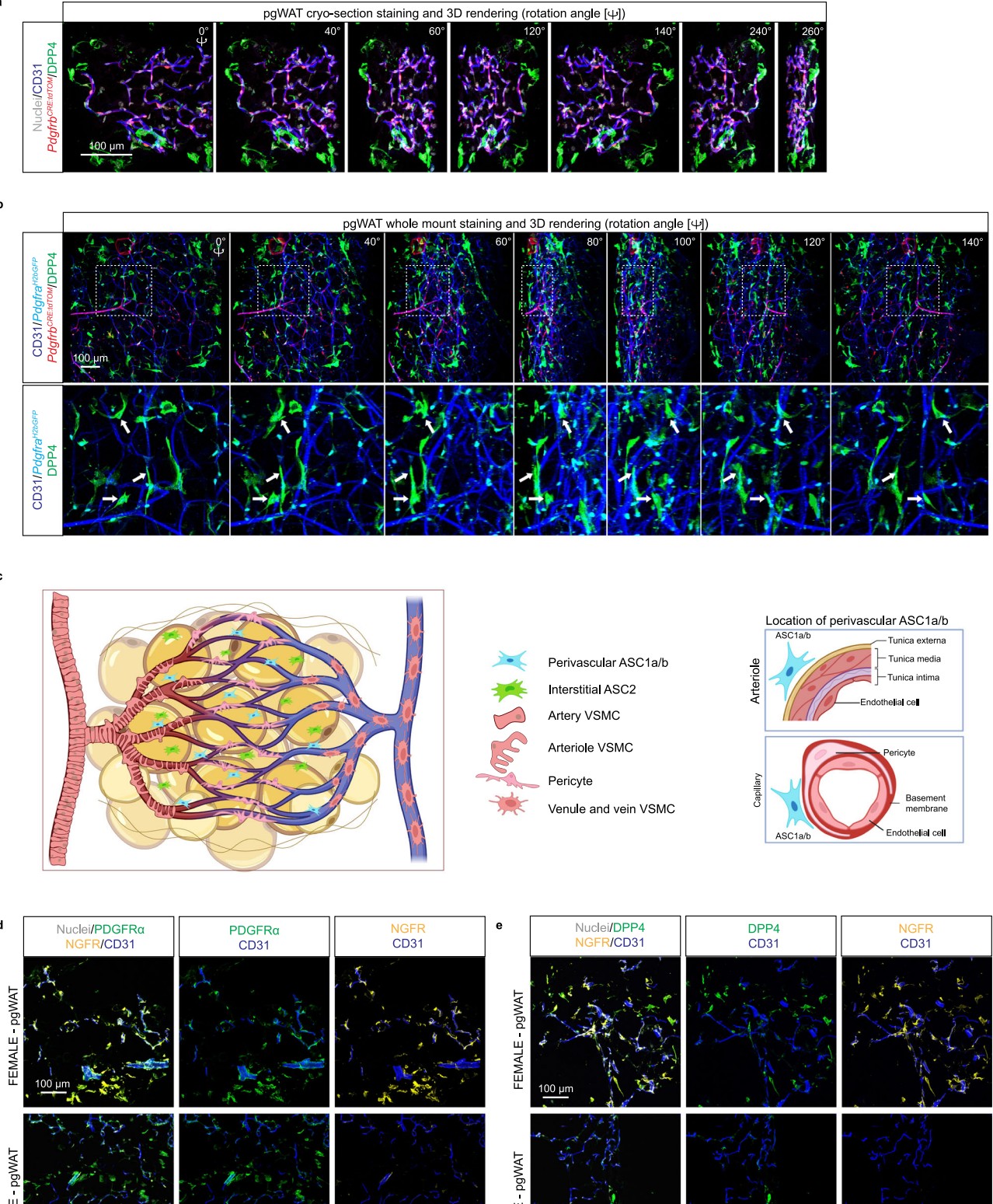

**Fig. 6 | Cell localization in perigonadal white adipose tissue in *Pdgfrb^CRE-tdTOM^/Pdgfra^H2bGFP^* reporter mice. a** pgWAT cryo-section staining and 3D rendering in *Pdgfrb^CRE-tdTOM^/Pdgfra^H2bGFP^* mice with anti-DPP4 antibody staining for ASC2. **b** pgWAT whole amount staining and 3D rendering in *Pdgfrb^CRE-tdTOM^/Pdgfra^H2bGFP^* mice with *Pdgfra* driven GFP expression marked in cyan color. **c** Schematic cartoon of our view of the perivascular and vascular cells in pgWAT. Created with BioRender.com released under a Creative Commons Attribution-NonCommercial-

NoDerivs 4.0 International license (https://creativecommons.org/licenses/by-nc-nd/4.0/deed.en). **d** Immunofluorescence staining of pgWAT from wilt-type mice for NGFR (encoded by the female ASC specific *Ngfr* transcript), PDGFRA and CD31 **e** same as in **d** but staining for ASC2 marker DPP4 instead of the ASC marker PDGFRA. ASC Adipose stem cells, GFP Green fluorescent protein, and pgWAT perigonadal White adipose tissue.

Lin⁻/CD142⁻/ICAM⁺ gating to isolate ASC1b, ASC2 and ASC1a populations respectively, and found that both ASC1a and ASC1b could be readily differentiated into mature adipocytes in vitro, and that the differentiation of ASC1b and ASC2 was inhibited by TGFβ, whereas ASC1a was not. Schwalie et al.[20] used a similar FACS-strategy to isolate ASC1b cells by dividing a Lin⁻/CD34⁺/SCA1⁺ fraction into either CD142 positive (ASC1b) or negative (ASC2 and ASC1a) cells, but in contrast to Merrick et al, their ASC1b (a.k.a. adipogenesis regulatory cells (Aregs)) demonstrated an inhibitory activity on adipogenesis and low adipogenic potential in vitro[20]. In our present analysis, DPP4⁻ (ASC1a/b) cells from male mice readily underwent adipogenesis in minimal adipogenic culture medium. A possible explanation for the different functional properties of ASC1b in the different studies (Supplementary Data 1, 2) is that the proportion of Aregs in the DPP4⁻ fraction may vary (it remains unknown in our data). In summary, our data confirm the intra-depot transcriptional heterogeneity of ASC suggested by others, but also uncovers differences in the functional properties of these cells in vitro that deserve further study.

It has long been observed that adipocytes arise in a perivascular niche of AT, which has inspired lineage tracing studies focused on blood vessel and associated cells. Our scRNA-seq data show that pgWAT ASC and mural cells are distinct. Surface marker profiles of cells in the WAT perivascular niche distinguish endothelial cells (CD31⁺), mural cells (PDGFRB⁺, PDGFRA⁻), ASC1 (PDGFRA⁺, DPP4⁻), and ASC2 (PDGFRA⁺, DPP4⁺) and their spatial distribution. In addition to the classical distinction between VSMCs around arteries and veins[72,73] and pericytes in capillaries (Fig. 5a, b), our data reveal transitional cell morphologies along the arterio-venous axis in WAT similar to those previously reported in the brain.

Previous work has shown that the DPP4⁺ ASC cells are excluded from the perivascular compartment and reside in the reticular WAT interstitium, a fluid-filled layer containing elastin and collagen fibers surrounding parenchymal cells in many organs[21]. Conversely, DPP4⁻ ASC1b cells have been suggested to reside in the perivascular space[20]. Accordingly, we find DPP4⁺ ASC2 in the pgWAT interstitial space without blood vessel association, whereas ASC1a/b are immediately outside of the mural cell coat. This location may allow regulation of vascular permeability and blood pressure in addition to serving as a pre-adipocyte niche[73]. The recent progress in AT scRNA-seq will likely bring further understanding of the signaling networks that regulate interactions between EC, mural cells, ASC, and mature adipocytes in health and obesity[74].

## Methods

### Reagents
For antibodies used for imaging see Supplementary Table 7. For antibodies used for FACS-sorting see Supplementary Table 8. For medium and enzymes used for the digestion of tissues see Supplementary Table 9. For key reagents for ASC proliferation, differentiation, and imaging see Supplementary Table 10. For Primers used for qPCR (Method: SYBR Green) see Supplementary Table 11.

### Animals
All mouse experiments were conducted according to local guidelines and regulations for animal welfare, experiments on reporter mice strains were covered by ethical permits approved by Linköping's animal Research Ethics, approval ID 729 and 3711-2020, whereas experiments with wild-type mice for in vitro studies and FACS bulk RNA-seq isolations were covered by ethical permits approved by Gothenburg's animal research Ethics committee, approval ID: 000832-2017. All animals were maintained on a 12 h light – 12 h dark cycle in a temperature-controlled environment (22 °C), with free access to water and chow-diet. For the scRNA-seq experiments and tissue imaging we used a Pdgfrb^GFP (Genesat.org, Tg(Pdgfrb-eGFP)) mouse strain that have been backcrossed to the C57BL6/J background (The Jackson Laboratory,

C57B16/J). Pdgfra^H2b-GFP (B6.Cg-Pdgfra^tm11(EGFP)Sor)[75] mice were crossed to Pdgfrb-Cre^ERT2 (Tg(Pdgfrb-CRE/ERT2)6096Rha)[76] and Ail4-TdTomato(B6.Cg-Gt(ROSA)26Sortm14 (CAG-TdTomato)Hze)[77] mice to generate tissue imaging as shown in Fig. 6. Pdgfrb-Cre^ERT2 was induced with 3 doses of tamoxifen (2 mg) in peanut oil by oral gavage at 4 weeks of age to activate TdTomato expression. For FACS-sorted ASC and mature adipocyte isolation used for generating bulk RNA-seq samples we used 16 C57BL6/J mice (eight males and eight females) of the age of 18 weeks (supplied by Charles River). For the castration/ovariectomy study, eight ovariectomized (study code: OVARIEX), eight castrated (Study code: CASTRATE) and aged matched controls of the strain C57BL/6 J were supplied from Charles River and terminated at 10 weeks of age. Castration/Ovariectomy was conducted at the age of four weeks. Animals were fasted for 4 h before termination for studies of bulk RNA-seq of FACS sorted ASC. For in vitro experiments (Fig. 4e, f) we used wild-type C57BL/6 N mouse strain (supplied by Charles River) and C57BL/6 J for the rest of the experiments displayed in Figure 4 and Supplementary Figs. 8, 9. Adult mice of both sexes were used at an age range of 12–20 weeks for scRNA-seq and in vitro experiments.

### Isolation of single cells from mouse adipose tissue for scRNA-seq and in vitro experiments
Mice were euthanized according to the ethical permission by cervical dislocation before inguinal/perigonadal white adipose tissue was removed and placed into cold PBS solution. The adipose tissue was then cut into smaller pieces before incubation in dissociation buffer (Skeletal Muscle dissociation kit, Miltenyi), supplemented with 1 mg/ml Collagenase type IV-S at 37 °C with horizontal shaking at 500–800 rpm. For all in vitro experiments a different enzymatic mixture was used with 2 mg/ml Dispase ii, 1 mg/ml Collagenase I, 1 mg/ml Collagenase II and 25 units/ml of DNAse dissolved in DMEM. The tissue was further disintegrated by pipetting every 10 min during the 30-minute-long incubation. The cell suspension was then sequentially passed through a 70 μm and 40 μm cell strainers, before 5 ml of DMEM was passed through both strainers as final washing step. Cells were then spun at 250xg for 5 min, the buffer was removed, and the pellet was re-suspended in FACS buffer (PBS, supplemented with 0.5% BSA, 2 mM EDTA, 25 mM HEPES). Cells were then labeled with fluorophore-conjugated antibodies (anti-CD31, anti-CD34, anti-DPP4, anti-CD45) for 20 min on ice, then centrifuged at 250xg for 5 min, after removal of the supernatant the pellet was re-suspended with FACS buffer and kept on ice. For isolation of mature adipocytes, the crude cell suspension was passed through a 100 μm instead, and the remaining cell suspension was left on ice for a few minutes allowing floating mature adipocytes to be collected from the surface of the suspension. The mature adipocytes were transferred to a separate Eppendorf tube where redundant cell suspension medium was removed with a syringe.

### Fluorescent activated cell sorting (FACS) for scRNA-seq
Cell suspension derived from Pdgfrb^GFP reporter mice were stained with antibodies and subjected to flow cytometry sorting as described previously[25]. Briefly. Beckson Dickson FACS Aria III or FACS Melody Cells instruments equipped with 100 μm nozzle were used for sorting cells into individual-wells of a 384 well-plate containing 2.3 μl of lysis buffer (0.2% Triton X-100, 2 U/ml RNAse inhibitor, 2 mM dNTPs, 1 μM Smart-dT30VN primers). Correct aiming was assured by test-spotting beads onto the plastic seal of each plate. Sample plates were kept at 4 C during sorting and directly placed on dry-ice afterwards, plates were stored in −80 °C until further processing. The gating-strategy was applied to enrich cells expressing protein signatures of interest but not used for cell identification. For FACS-sorting of single cells: First, a gate of forward and side scatter area (FSC-A/SSC-A) on the linear scale was set generously in order to only eliminate cells with low values (red blood cells and cell debris), a second gate for double discrimination

was used based on distance from the diagonal line in the FSC-A/FSC-height plot, the third selection criteria was based on fluorescent signaling, with "fluorescent minus one" or mice negative for the GFP-reporter used as gating controls. Cells negative for CD45-staining were first selected, further gating were then either based on Pdgfrb$^{GFP+}$/CD31 + , CD31-/ Pdgfrb$^{GFP+}$ or CD31-/ Pdgfrb$^{GFP+}$ / DPP4± selections.

## RNA isolation and quantification of differentiated adipose stem cells

Qiagen's RNAeasy plus micro kit (Cat. No. 74 034) was used for RNA isolation from in vitro differentiated ASCs The High-Capacity cDNA Reverse Transcription kit (Cat. No. 4368814) was used to generate cDNA from RNA, and SYBR Green PCR Master Mix (Cat. No. 4309155) with custom primers from Thermo Fisher Scientific were used for relative quantification of mRNA levels (see separate Supplementary table for primer sequences). The experiments were repeated three times using in total 4 mice (two females and two males).

## Smartseq2 library preparation and sequencing

Isolation of mRNA molecules from single cells with subsequent cDNA synthesis and sequencing was carried out as described previously[22,25]. Briefly, cDNA was synthesized from mRNA using oligo-dT primers and SuperScript II reverse transcriptase (ThermoFischer Scientific). Templated switching oligo (TSO) was used for synthesizing the second strand of cDNA before amplification by 23-26 polymerase chain reaction (PCA) cycles. Purified amplicons were then quality controlled using an Agilent 2100 Bioanalyzer with a DNA High sensitivity chip (Agilent Biotechnologies). QC-passed cDNA libraries were then fragmented and tagged (tagmentation) using Tn5 transposase, and samples from each well were then uniquely indexed using Illumina Nextera XT index kits (set A-D). The uniquely labeled cDNA libraries from one 384-well plate were then pooled to one sample before loaded onto one lane of a HiSeq3000 sequencer (Illumina). Dual indexing and single 50 base-pair reads were used during sequencing.

Qiagen's RNeasy Micro kit (Cat. No. 74004) was used for the isolation of RNA from FACS-sorted ASC cells (5000 cells per sample) for bulk smartseq2 library preparation. For isolation of RNA from mature adipocytes, QIAzol lysis reagent (Cat. No. 79306) was first used followed by the addition of chloroform and subsequent integration of the RNA containing aqueous phase with the workflow of the RNeasy clean up. RNA samples were diluted to 3 ng/μl and 5 ng of RNA was used for cDNA synthesis according to the smartseq2 protocol. For FACS-sorted ASC bulk samples, RNA from approximately 300 cells were used for cDNA synthesis, and 16 PCA-cycles were used for cDNA amplification for both FACS-sorted ASC and bulk mature adipocyte samples. Samples derived from mature adipocytes were sequenced in technical duplicates, whereas FACS-sorted ASC bulk samples were sequenced in triplicates. The average quantified raw read counts were then calculated for the technical replicates prior to DE-analysis.

## Smartseq3 library preparation and sequencing

RNA samples were extracted using Qiagen's RNAeasy plus micro kit (Cat. No. 74 034) from 32 FACS sorted adipose stem cells from castrated male ($n = 8$), ovariectomized female ($n = 8$), female ($n = 8$) and male ($n = 8$) control mice, and 29 in vitro cultivated crude SVF cells that had been proliferated for 4 days in PM1-medium supplemented with 1 nM basic FGF (same procedure as for in vitro differentiation protocols), the RNA concentration were normalized to 3 ng/ul in 20 μL Nuclease-free water. For library construction and sequencing strategy, we adopted the sensitive smart-seq3 protocol to perform our bulk RNA-seq in single-cell format[78], with some modifications as follows. Two μL of each RNA sample was transferred into one well of 384-well plate, where contained 0,3 μL 1% Triton X-100, 0,5 μL PEG 8000 40%, 0,04 μL RNase inhibitor (40U/μL), 0,08 μL dNTPs mix (25 mM) and 0,02 μL Smart-dT30VN/dT (100 μM, 5'-ACGAGCATCAGCAGCATACGA

TTTTTTTTTTTTTTTTTTTTTTTTTTTTTTTVN-3'). Reverse transcription was performed after mixture with 1 μL/well buffer (0,10 μL Tris-HCl, pH 8.3, 1 M, 0,12 μL 1 M NaCl, 0,10 μL MgCl$_2$, 100 mM, 0,04 μL GTP, 100 mM, 0,32 μL DTT, 100 mM, 0,05 μL RNase Inhibitor, 40 U/μl, 0,04 μL Maxima H Minus RT, 200 U/μL, 0,08 μL SmartSeq3 TSO, 100 μM, 5'-AGAGACAGATTGCGCAATGNNNNNNNNrGrGrG-3' and 0,15 μL H$_2$O, where N indicates random sequence, while r denotes RNA characteristics). The PCR program was 3 min at 85 °C, 90 min at 42 °C, 10 cycles of 2 min at 50 °C and 2 min at 42 °C, followed by 85 °C for 5 min and incubation at 4 °C. Immediately after the reverse transcription, PCR was performed after mixture with 6 μL/well of PCR buffer (2 μL 5X KAPA HiFi HotStart buffer, 0,12 μL 25 mM dNTPs mix, 0,05 μL 100 mM MgCl$_2$, 0,2 μL KAPA HiFi HotStart DNA Polymerase (1 U/μL), 0,05 μL Forward PCR primer (100 μM, 5'-TCGTCGGCAGCGTCA-GATGTGTATAAGAGACAGATTGCGCAA*T*G-3', * = phosphothioate bond), 0,01 μL Reverse PCR primer (100 μM, 5'-ACGAGCATCAGCAG-CATAC*G*A −3', * = phosphothioate bond) and 3,57 μL H$_2$O. The amplification program was: initial denaturation at 98 °C for 3 min, N cycles of denaturation at 98 °C for 20 sec, annealing at 65 °C for 30 sec and elongation at 72 °C for 4 min, followed by a final elongation at 72 °C for 5 min and incubation at 4 °C.

The cDNA was purified using 6 μl/well SeraMag beads (containing 17% PEG) and eluted into 10 μl/well elution buffer according to the user manual. After quality control using Bioanalyzer 2100 (Agilent), the cDNA was diluted to 200 pg/μl and combined into one 384-plate for tagmentation. For each sample, 1 μL diluted cDNA was mixed with 925 nL tagmentation buffer (containing 20 nL Tris-HCL pH 7.5 1 M, 100.5 nL MgCl$_2$ 100 mM, 100.5 nL Dimethylformamide (DMF) and 704 nL H$_2$O) and 75 nL Tn5 enzyme, and subjected to incubation at 55 °C for 10 min. Then, the reaction was immediately terminated by a mixture with 500 nL 0.2% SDS and incubation at room temperature for 5 min. Subsequently, 1 μL index combination (500 nL for each) and 4.4 μL PCR mix (1,40 μL 5x Phusion HF buffer, 0,06 μL dNTPs mix (25 mM), 0,04 μL Phusion HF (2U/μl) and 2,90 μL water) were applied to each well for enrichment PCR using the following program: gap-filling at 72 °C for 3 min, initial denaturation at 98 °C for 3 min, 13 cycles of denaturation at 98 °C for 10 sec, annealing at 55 °C for 30 sec and elongation at 72 °C for 30 sec, followed by a final elongation at 72 °C for 5 min and incubation at 4 °C.

In the end, the libraries were pooled for purification using a two-step purification protocol. First, 24% SeraMag beads were mixed with libraries at a volume ratio of 0.6:1, followed by successive 8 min incubation without and with a magnet stand. After removal of the supernatant, the beads were washed with 80% ethanol and eluted into 50 μl elution buffer. Then, the 50 μl elute was thoroughly mixed with 50 μL H$_2$O and 70 μl SPRI beads, followed by successive 2 min incubation without and with a magnet stand. After removal of the supernatant, the beads were washed with 85% ethanol and eluted into 25 μl elution buffer. Finally, the purified library pool was quality-controlled and diluted for sequencing at Novaseq 6000 (Illumina).

## In vitro differentiation of adipose stem cells

Stromal vascular cells from both iWAT and pgWAT were isolated according to the procedure described above. Thereafter cells were labeled with four fluorophore-conjugated antibodies (anti-CD45, anti-CD31, anti-CD34, and anti-DPP4) for 30 min on ice. The cell suspension was then centrifuged at 300 g for 5 min, supernatant removed, and pellet re-suspended in FACS-buffer. Cells were then loaded into a SH800 Sony cell sorter, and two adipose stem cell populations, CD45-/CD31-/CD34 + /DPP4+ and CD45-/CD31-/CD34 + /DPP4-, were gated and selected for sorting. Fluorescent minus one controls were used for ensuring correct gating. Cells were collected in PM-1 medium (Zenbio), supplemented with 1 nM of basic FGF, and seeded in 96-well plates. DPP4+ and DPP4-negative cells were seeded at 10,000 cells and 12 500 cells per well, respectively. DPP4-positive cells were seeded at a lower

density since they proliferate faster than the DPP4- population. For the experiment with crude SVF cells, 15-20 000 cells were seeded. After becoming confluent after 3-4 days of proliferation, the differentiation was initiated by changing medium to basal medium (Zenbio) supplemented with 3% FBS, 1% pen/strep, 0.5 mM IBMX, 1 µM dexamethasone, and 100 nM of insulin. The medium was then changed after 48 h to a maintenance medium including BM-1, 3% FBS, 1% Pen/strep, 1 µM of pioglitazone, and 100 nM insulin with medium changed every other day (pioglitazone was not included in the differentiation experiments with AngII). A second treatment group with cells subjected to only 100 nM of insulin in basal medium (Zenbio) supplemented with 3% FBS, 1% pen/strep, was also included in the studies. After 8 days of differentiation, cells were stained with propidium iodine, Bodipy, and Hoechst 33342.

Cells were then imaged with an ImageXpress widefield fluorescence microscope using the 4x objective (10x objective for crude SVF cells). Images were analyzed with the MetaXpress software applying the multi-wavelength cell scoring application module, counting cells (Hoechst positive), lipids (Bodipy positive) and the number of dead cells (propidium iodine positive) for viability measurements; for more detailed settings for the quantification of the differentiation of FACS sorted ASCs see Supplementary table 12.

Cells that were positive for both Bodipy and Hoechst were determined as differentiated (the number of W2 positive cells). The percentage of differentiated cells were then calculated by dividing the amount of lipid-filled cells by the total amount of cells in the well. Wells with a viability below 85% were not included in the analysis. The experiment was repeated four and five times, for crude SVF and FACs sorted ASC, respectively, with the average results from the technical well-replicates presented in figures. Experiments with AngII were carried out five times and each plate had three and six technical replicates for treatment and control wells, respectively.

## In vitro proliferation rate assay of adipose stem cells

Cell proliferation rates were measured on two stem cell populations, CD45-/CD31-/CD34 + /DPP4- (ASC1) and CD45-/CD31-/CD34 + /DPP4+ (ASC2). Briefly, cells were isolated from inguinal WAT from adult mice and subjected to FACS-sorting as previously described. Cells were sorted at a density of 7500 cells per well in a 96-well plate in PM-1 (ZenBio) medium supplemented with 1 nM of basic FGF. Cell proliferation was then measured by analyzing the level of confluence using the IncuCyte S3 live-cell analysis system. Images were taken every fourth hour for 120 h, with the 10x objective. This experiment was repeated three times with similar results.

## Explant lipolysis assay

Aged-matched mouse (Age:12-14 weeks, strain: C57BL6/J) were fasted for 3 h before Inguinal white adipose tissue was removed and put into ice-cold PBS without $Ca^{2+}/Mg^{2+}$. The Fat pads were then cut into smaller pieces before 25-30 mg of tissue was put into 100 ul of KREBS ringer buffer (,25 mM HEPES, 120 mM NaCl, 10 mM $NaHCO_3$,, 4 mM $KH_2PO_4$, 1 mM $MgSO_4$, 0.75 mM $CaCl_2$) with 2% fatty acid free BSA, 5 mM, 5 mM Glucose. The levels of released non esterified fatty acids (NEFA) in the medium were then measured after 4 h in a 37 C incubator (5% $CO_2$). Each condition had 8 technical replicates and the average value from the replicates is presented in Fig. 3d. The experiment was repeated three times using two males and two females mice each time, fat pads from mice with the same sex were pooled. NEFAs were analyzed using an ABX Pentra 400 instrument (Horiba Medical, Irvine, California, USA) and concentrations were determined by colorimetry with Fujifilm NEFA-HR(2) (ref 43491795 (R1) and 436-91995 (R2)). Fujifilm NEFA standard (Ref 27077000) was used as calibrator and Seronorm™ Lipid (Ref# 100205, Sero AS, Billingstad, Norway) was used as control.

## Immunofluorescence

Cryo-sections: Standard protocols for immunostaining were applied. In brief, adipose tissues were harvested from euthanized mice as described above and immersed in 4% formaldehyde solution (Histolab) at 4 °C for 4-12 h. Thereafter, the tissues were transferred to 20-30% sucrose/PBS solution at 4 °C for at least 24 h. For cryo-sectioning, the tissues were embedded into cryo-medium (NEG50) and sectioned at a CryoStat NX70 (ThermoFisherScientific) into 14 – 50 µm thick sections, collected on SuperFrost Plus glass slides (Metzler Gläser) and stored at −80 °C until further processing. Of note, for sectioning of adipose tissues the biopsy and knife of the cryostat were cooled down to at least −30 °C. For staining, the tissue sections were allowed to dry at RT for about 15 min and were briefly washed with PBS. Thereafter, the sections were treated with blocking buffer (Serum-free protein blocking solution, DAKO) supplemented with 0.2% Triton X-100 (Sigma Aldrich). Then, the tissue sections were incubated with primary antibodies, diluted in blocking buffer supplemented with 0.2% Triton X-100 over night at 4 °C. Followed by a brief wash with PBS-T (PBS supplemented with 0.1% Tween-20) and incubation with fluorescently conjugated secondary antibodies diluted in blocking buffer at RT for 1 h. Primary and secondary antibodies were used according to the manufacturers' recommendations (see Supplementary table 7). For nuclear (DNA) stain, Hoechst 33342 was used at 10 µg/ml together with the secondary antibodies. Sections were mounted with ProLong Gold mounting medium (ThermoFisher Scientific). Micrographs were acquired using a Leica TCS SP8 confocal microscope with LAS X software (version: 3.5.7.23225, Leica Microsystems) and graphically processed and adjusted individually for brightness and contrast using ImageJ/FIJI software[79] for optimal visualization. All images are presented as maximum-intensity projections of acquired z-stacks covering the thickness of the section.

Whole mount: Adipose tissues were harvested and processed as described above. After fixation and sucrose treatment (see above), small pieces of less than 1 mm thickness were cut and washed in PBS-T buffer at RT for 6-8 h with end-over-end rotation. Thereafter, the tissues were transferred into blocking buffer supplemented with 0.5% Triton X-100 for over-night incubation at 4 °C with end-over-end rotation. Primary antibodies were diluted in blocking buffer supplemented with 0.5% Triton X-100 and incubated with the tissues for 72–96 h at 4 °C with end-over-end rotation. Thereafter, the tissues were washed with PBS-T for 6–8 h at 4 °C with end-over-end rotation. Secondary antibodies were diluted in blocking buffer, supplemented with 0.5% Triton X-100 and 10 µg/ml Hoechst 33342, and incubated with tissues at 4 °C overnight with end-over-end rotation. Before mounting, tissues were washed with PBS-T for 6–8 h at 4 °C and then mounted on Leica frame slides (1.4 µm PET, Leica Microsystems) using ProLong Gold mounting medium. Micrographs were acquired using a Leica TCS SP8 confocal microscope and graphically handled as described above.

## Raw sequence data processing Smartseq2 protocol

Single-cell cDNA library samples from one 384-well plates were pooled and sequenced on a HiSeq 3000 sequencer (Illumina), with one flow-cell lane per plate. In total 11 plates of cells from eight (five females and three males) mice were used for this study. The samples were then analyzed using standard parameters of the illumina pipeline (bcl2fastq) using Nextera index parameters. Individual fastq-files were mapped to the mouse reference genome (mm10-build94) with the STAR aligner, and raw reads for each gene was quantified using Salmon. As technical controls, 92 ERCC RNAs were spiked in the lysis buffer and included in the mapping. Raw read counts were then imported into R with the tximport-package and combined into one expression matrix showing raw counts per gene for each single cell. The R package biomaRt was used to convert ensemble ids to gene names, locate genomic location and gene biotype.

The SingleCellExperiment package in R was then applied for downstream processing of the expression matrix. First, cells with fewer than 150,000 total reads and more than 5,000,000 reads were filtered out. Cells that had fewer than 1000 genes expressed and that had a high percent of the reads mapped to the mitochondrial genome (>17.5%) or to the ERCCs (>20%) were removed from the dataset. Additionally, genes that had less than 10 reads in no more than three cells were removed. A final filter step was added by applying the gene.vs.molecule.cell.filtering function in the pathway and gene set over-dispersion analysis (Pagoda2) package, removing cells that were determined to be outliers in their gene vs total counts ratio.

For bulk RNA-seq samples with fewer than 250 000 total reads were filtered out. Samples that had fewer than 8000 genes expressed and that had a high percent of the reads mapped to the mitochondrial genome (>15%) or to the ERCCs (>1%) were removed from the dataset.

The Seurat-package was then applied to perform principal component analysis with RunPCA function using variance-stabilizing transformation as a selection method for finding the 3000 most variable genes. The clustering of cells was performed by first applying the findNeighbors function using 14 PC:s dimensions followed by FindClusters (resolution=1.1) function. For dimensional reduction visualization, UMAP projection was applied using the Seurat package, the top 2000 over dispersed genes were used as input variable. The few cells projected in connection to other clusters in UMAP compared to most cells in its cluster were removed, since this indicated contamination of other cell classes in those samples. Before removal, the contamination in these specific samples were verified by calculating a ratio between the percentage of read counts (of total read counts in the sample) belonging to marker genes to cells in its close surroundings compared to marker genes for its cluster. For example, if an endothelial cell was projected into pericytes population in the UMAP, a ratio of the percentage pericytes markers divided by the percentage of endothelial cells marker genes were calculated. This ratio was significantly higher in all the endothelial-placed cells as compared to cells located in the area of the pericyte cluster. A second round of clustering after this cleaning step was performed as described above.

### Raw sequence data processing Smartseq3 protocol

Raw fastq-files were collected, sequencing adapters were then trimmed from the remaining libraries using NGmerge (v0.3)[80] and read quality for all libraries was assessed using FastQC (v0.11.9)[81], Qualimap (v2.2.2d)[82] and samtools stats (v1.15)[83]. Quality control (QC) metrics for Qualimap and samtools were based on a STAR (v2.7.10a)[84] alignment against the mouse genome (GRCm39, Gencode vM32). UMI information was evaluated with UMI tools[85]. Next, QC metrics were summarized using MultiQC (v1.12)[86]. A mouse transcriptome index consisting of cDNA and ncRNA entries from Gencode (vM32) was generated and reads were mapped to the index and quantified using Salmon (v1.9.0)[87]. The bioinformatics workflow was organized using Nextflow workflow management system (v22.04.5)[88] and Bioconda software management tool[89].

The raw count matrix including the total reads from both UMI and non-UMI containing sequences was imported into R for further downstream processing. This included removing genes that had less than 5 reads in no more than three cells. Samples that had less than 8500 genes detected in FACS-sorted ASC samples (castration/ovariectomy study) were removed, whereas samples in the in vitro cultivated crude SVF group with less than 10,500 genes detected were removed. Overall, the number of samples that passed this criterion from the castration/ovariectomy study were 58, including eight samples for all groups except ASC1 cells from male control (n = 7) and female ovariectomized mice (n = 3). The average total read count was 690,000 reads for these samples. The number of samples that passed this criterion from the in vitro cultivated crude SVF study were 29, including nine samples in both male and female iWAT groups, six

samples for male pgWAT and five samples for female pgWAT. The average total read count was 620 000 reads for these samples. Of note, low read counts from Y-chromosome genes Ddx3y (<30 counts) and Eif2s3y (<20 counts) were detected in female control samples, which indicates weak contamination of male mRNA, however, relative low counts of the male specific genes Sult1e1 and C7 in comparison the male control samples suggest that the samples are "clean" and the results can be trusted. Low levels of Ddx3y and Eif2s3y (<10 counts) is also detected in in vitro cultivated crude SVF group from female pgWAT samples, this low grad of contamination most likely did not impact the conclusion made in this paper.

### Differential expression and pathway analysis

For pathway analysis of transcriptomic data, QIAGEN's Ingenuity Pathway Analysis (IPA) application was used. All presented canonical pathways, diseases, and molecular functions displayed in this paper were significantly enriched, the threshold for adjusted p-value was set to below 0.05. The list of genes used as input for the IPA application were derived according to the differential expression analysis described below.

For differential expression analysis, the pseudo bulk EdgeR-LRT method was used (R-package: edgeR v:3.22.5) with raw counts as input. A gene was classified as significantly differentially expressed if it generated a Benjamini-Hochberg adjusted p-value for multiple testing below 0.05 and if it had a raw read count above 600 reads in at least 2 pseudo samples with a fold change of more than 2.6. These settings were used for generating differentially expressed genes between male and female ASC, the pseudo bulk method grouped cells from the same mice in each sex, resulting in n = 3 for male cells and n = 4 for female cells. The Seurat FindMarkers function was applied using the Wilcoxon rank sum test with raw read counts as input for generating marker genes for clusters and between male and female EC cells. A gene was classified as significantly differentially expressed if it generated a bonferroni adjusted p-value for multiple testing below 0.05 and if it was expressed by at least half of the cells in the group with a fold change of more than 2.6 (min.pct = 0.5, logfc.threshold = 1.4). Slightly different settings were used for generating DE-genes specific for adipose ASC in comparison to heart and skeletal muscle fibroblasts (min.pct = 0.25, logfc.threshold = 0.7). For FACS-sorted bulk RNA-seq samples, DE-genes between male and female ASC cells were calculated using DESeq2, a gene was classified as significantly differentially expressed if it generated an adjusted p-value for multiple testing below 0.05 and if was expressed by at least half of the samples. Both DPP4+ and DPP4- cell populations were used for this analysis. For pgWAT, 14 samples in both males and females passed the filtration criteria mentioned previously (seven DPP4+ and seven DPP4-), whereas for iWAT, the female and male group consisted of 14 samples (seven DPP4+ and seven DPP4-) and 15 samples (seven DPP4+ and eight DPP4-), respectively. The gene Gm20400 was also sexually dimorphic genes under these settings however the gene was not included in Fig. 2c since it is a long non-coding RNA and there is limited knowledge of its function.

For FACS-sorted bulk RNA-seq samples from the castration/ovariectomy study, DE-genes between male and female ASC cells from iWAT were calculated with DESeq2, for this analysis the sexually dimorphic genes that were DE in the initial ACS-sorted bulk RNA-seq from iWAT/scRNA-seq comparison (36 + 4 genes, Fig. 2c) were only analyzed. A gene was classified as significantly differentially expressed if it generated an adjusted p-value for multiple testing below 0.05 and if it had fold change difference of at least 2 in the comparison of the control groups. To validate if the a gene was impacted by sex hormones, it also needed to show no statistically significant difference in the DE-analysis between the castration-female control/ovariectomy-male control comparisons.

For bulk RNA-seq samples of cultivated crude SVF cells (Supplementary Fig. 8), a gene was defined as enriched in either pgWAT or iWAT samples if the DE-analysis using Deseq2 generated a p-value below 0.05 or enriched in female pgWAT samples compared to a group of iWAT samples and male pgWAT. See source data for Supplementary Fig. 8g and k.

## Other bioinformatic analyses

Pearson's r values were calculated using the cor function in R stats package with the scaled average expression values as input variables (AverageExpression function in Seurat was applied) for marker genes or genes of a specific genotype if indicated. The R package corrplot was used to visualize the results and the groups were order according to hierarchical clustering method "complete" with blue lines displaying the results of that clustering. Dotplots were generated using Seurat's DotPlot function using normalized values (normalized method: "LogNormalize", scale.factor =500 000) as input. For data downloaded from external source the log normalized values in the provided R-object were used. The scaling function in DotPlot function was turned on, resulting in scaling of the average log normalized values, this means that the scaled values will always be plus or minus 0.7 (square root (2) / 2) for all comparisons between two groups. The group with the highest expression will have a value of 0.7 and the group with the lowest expression will have a value of −0.7. This also means that the color intensity indicating the size of fold difference is misleading, it is therefore better to view the result as an indicator of which of the two groups has the highest average expression. The statistically significant enriched *Hox* genes in our scRNA-seq data with a fold change of above 2.6 in females are *Hoxa9, Hoxa10, Hoxc10, Hoxa11os* and *Hoxa11*, and for males *Hoxb5, Hoxc5, Hoxc6* and *Hoxc8*.

For validation of mouse data, we used human single nuclei RNA sequence data published by Emont, MP et al.[41]. The human adipose single-nucleus raw count data (10x chromium-v3) and metadata were downloaed from the Broad Institute's single cell portal webpage (link: https://singlecell.broadinstitute.org/single_cell/study/SCP1376/a-single-cell-atlas-of-human-and-mouse-white-adipose-tissue). Both "human_ASPCs.rds" and "human_adipocytes.rds" were used for our analysis and the data was generated from subcutaneous (subc) adipose tissue from ten female and three male donors, for visceral (visc) adipose tissues the data was derived from seven female and three male donors. The number of cells per cell type and fat depot from Emont et al.[41] is presented in Supplementary Table 6.

For comparison of ASC to fibroblast identified in heart and skeletal muscle raw fastq-file were provided by Muhl L et al.[25], and raw sequence processing was done as described above. The number of cells per cluster from Muhl L et al.[25] is presented in Supplementary Table 2.

For comparison to the Tabula Muris data[38], the "facs_Fat_seurat_tiss.Robj" file was downloaded from the human cell atlas data portal (link: https://data.humancellatlas.org/explore/projects/e0009214-c0a0-4a7b-96e2-d6a83e966ce0/project-matrices). Before generating scaled dotplots in Fig. 2, genes that had less than 10 reads in no more than three cells were removed from the countmatrix of each fat depots MSC and raw counts were log normalized to the total counts in each cell, using the NormalizeData function in Seurat. The number of cells per adipose depot and sex from Tabula Muris[38] is presented in Supplementary Table 5.

For comparison to fibroblasts from Buechler M.B et al.[29], the mouse steady-state atlas was used, data was downloaded from https://www.fibroxplorer.com/download. The original source from which the data was derived can be seen in the Supplementary Table. The number of cells per tissue from Buechler et al.[29] is presented in Supplementary Table 3.

For our scRNA-seq data from pgWAT the sex and animal metadata are displayed in Supplementary Table 1.

## Statistics and reproducibility

Barplots for scRNA-seq and bulk RNA-seq data displays mean ± SEM of the raw read counts. Figure 4d displays the mean ± SEM and the barplots in Fig. 4e displays the mean ± standard deviation. In Fig. 4e, the number of dots represents the number of biological replicates, and for the gene expression data in Fig. 4d, each dot represents a technical replicate derived from three experiments. Statistics in Fig. 3d, e were calculated with two-way ANOVA using Sidak's multiple comparisons test. Statistics in Fig. 4b were calculated with a two-sided unpaired t-test (Prism) for the data points collected at the final time point, in Fig. 4d, e statistics were calculated with a two-way ANOVA and Mixed-effects analysis, respectively, using Tukey's multiple comparison test (Prism). For Supplementary Fig. 8a, statistics were calculated with a two-way ANOVA, using Tukey's multiple comparison test (Prism). Adjusted P-values for multiple testing were used, ($*P < 0.0332$, $**P < 0.0021$, $***P < 0.0002$, and $****P < 0.0001$). The statistics in Fig. 4d were based on the delta Ct-values using *TBP* as a house keeping gene. Regarding imaging, if not further specified, all antibody immunofluorescence experiments have been performed at least twice using identical or varying combinations of antibodies, obtaining similar results from tissue samples of at least two individual mice. The whole mount staining experiments were performed twice, analyzing tissue samples from two individual mice. For validation of sex-specific expression of NGFR, additionally two female and two male littermates were analyzed. In Fig. 4c, representative images of the level of differentiation is displayed and similar results have been repeated in at least three independent experiments.

## Reporting summary

Further information on research design is available in the Nature Portfolio Reporting Summary linked to this article.

## Data availability

The RNA-seq raw data generated in this study have been deposited in the NCBI's Gene Expression Omnibus database under accession code GSE273393 (scRNA-seq), GSE273413 (FACS_pgWAT_iWAT_ASC), GSE273407 (FACS_castovary_ASC), GSE272408 (bulk_adipocytes) and GSE273416 (in vitro_SVF). The scRNA-seq and bulk RNA-seq data of FACS sorted ASC are available as a searchable database at https://betsholtzlab.org/Publications/WATstromalVascular/database.html. Source data are provided with this paper.

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

## Acknowledgements

We would like to acknowledge the staff at the Single Cell Core Facility (SICOF) at Karolinska Institute and at the animal facilities at both Karolinska Institute and AstraZeneca for their work. We would also like to acknowledge the funding support from AstraZeneca.

## Author contributions

M.U. was responsible for hypothesis generation, conceptual design, experiment design and performance, data analysis, and manuscript preparation. L.M. experiment design and performance, data analysis, and manuscript preparation. G.G. was responsible for experiment design and performance. J.L. was responsible for data generation, experiment design, and performance. H.P. was responsible for data analysis, experiment design, and performance. I.A. was responsible for experiment design and performance. F.K. was responsible for data analysis. A.X.Z. was responsible for experiment design and performance. S.L. was responsible for experiment design and performance. S.G. was responsible for data generation, experiment design, and performance. B.B. was responsible for data generation, experiment design, and performance. K.P. was responsible for experiment design and performance. I.A. was responsible for experiment design and performance. D.K. was responsible for experiment design and performance. L.A. was responsible for experiment design and performance. L.H. was responsible for data curation and data analysis. M.J. was responsible for the experiment design. C.B. carried out supervision of work, hypothesis generation, conceptual design, data analysis, and manuscript preparation. X.R.P. carried out supervision of work, hypothesis generation, conceptual design, data analysis, and manuscript preparation.

## Funding

## Competing interests
The authors declare no competing interests
