## [Peer Review File · Nature Communications]

REVIEWER COMMENTS

Reviewer #1 (Remarks to the Author):

This well written manuscript describes the transcriptomes at a single-cell level of CD45- cells from the inguinal adipose depot of Pfgfrb-GFP transgenic mice. Analysis is complemented by comparison of iWAT fibroblast transcriptomes with existing datasets from heart and muscle, findings that iWAT fibroblasts are enriched in genes encoding for extracellular matrix. Variation due to sex, including Hox gene expression, are also observed. The authors confirm previously reported variation in in-vitro differentiation capacity related to cell surface expression of DPP4. In addition, the authors perform high quality immunofluorescence analysis of diverse progenitor populations in iWAT, at a single time point, finding varied localizations relative to the vasculature. In general, the results are very high quality, clearly presented, and consistent with the authors conclusions. However, there is little mechanistic or conceptual advancement over multiple already available studies of adipose progenitor cells in mice.

Specific comments:

1. The authors state that iWAT fibroblasts differ from muscle fibroblasts due to expression of adipocyte specific genes such as adiponectin and perilipin. These genes are indicative features of pre-adipocytes, such that cells expressing them should not be categorized as fibroblasts.
2. Similarities between CD45- cells of adipose tissue and muscle could be attributed to the presence of adipocytes amongst muscle fibres.

Reviewer #2 (Remarks to the Author):

This work by Uhrbom et al uses single cell RNA sequencing to study adipose stem cell populations and examine the differences in populations across sex. Their major findings are:

- There are multiple populations of fibroblasts and mural cells in the adipose tissue.
- There is sexual dimorphism in gene expression and some functionality of adipose stem cells.
- Detailed imaging of adipose tissue shows that some ASC and pericyte populations are located near the adipose tissue vasculature.

Overall, the questions here are interesting, and the authors provide a valuable resource in their deeper sequencing of adipose stem cells using smart-seq. However, the paper is disjointed and almost feels like two studies (one on sexual dimorphism in ASCs and one on imaging of the adipose

tissue vasculature) were merged into one paper. Additionally, a number of the findings they present (namely points 1 and 3 above) have already been made in other works, something the authors do not mention at all earlier in the work, presenting the findings as if they are novel and in one case describing previous literature as “scant”, but then the discussion section almost reads like a review article of previous work that their work now agrees with. This paper would be improved by focusing on the novelty in the paper (the characterization of the sexual dimorphism) with more functional assays. Specific comments are as follows:

1. The pathway analysis in figure 3 suggests different pathways may be active in male vs female ASCs. Some functional characterization of these pathways in ASCs from both sexes would enhance the paper, such as Seahorse or Clark electrode measurement of OxPhos.
2. When the authors differentiate male and female preadipocytes in culture, do they observe sexually dimorphic gene expression in the mature adipocytes? If so, how similar is it to what they see in the preadipocytes?
3. It is interesting that the authors see a sexually dimorphic difference in cultured ASCs from male and female animals, and raises the question of what differences are preserved in culture and what are not. If ASCs from male and female animals are cultured for say a week, do they still see the transcriptional differences that they see in freshly isolated ASCs?
4. The cells in clusters 9 and 13 were correctly removed by the authors in the analysis done in figures 1 and 2 because of high mitochondrial gene expression, which is indicative of low-quality cells. Why then did they re-include these cells for the analysis in figure 3? These cells should be removed from all analyses.
5. The authors describe the current literature on adipose tissue microvasculature as “scant”, yet there have actually been a number of papers published on the subject, including one also showing that pdgfrb+ cells are located near the vasculature (PMID 26626462) and one that shows that DPP4+ ASCs are also located near the vasculature (PMID 36168895). The authors should significantly modify their discussion of this data to include some of the large body of work that has been done to study the adipose tissue vasculature.
6. The inclusion of figures 5 and 6 in this paper are confusing, as they are not related to the sexual dimorphism of adipose precursor cells that the rest of the paper is focused on. It is especially confusing to see a large diagram in figure 6c that is only related to the last two figures of the paper and not a summary of the entire paper. Is there sexual dimorphism in any of the findings shown here?

7. The authors describe in detail removing pericytes with suspected endothelial cell transcript contamination but this could be the result of ambient RNA, a package designed to computationally remove ambient RNA like cellbender should be used on the data to see if this allows them to remove potentially contaminating transcripts from their dataset.

8. The utility of most of the data in figure 1 (the dotplot in panel c, panels d&f) is a little unclear. There has been extensive work already on characterizing adipose fibroblasts at single cell resolution (for example, see PMID 33981032 for comparison between adipose and other fibroblasts, see PMID 36889280 for a review of adipose fibroblasts), and so it's a little confusing to find things like enriched genes and pathways in adipose fibroblasts compared to other fibroblasts presented in a main figure like this. I appreciate that the authors want to pull in data from their previous work and don't have a problem with the analysis itself, but the way this data is presented ignores a lot of previous work on the subject. The discussion does have almost a review of some of the previous work but the authors should also modify the results section to better place their work in the context of previous work.

9. Currently the labeling of ASCs varies widely by study, but there has been some effort to create consensus ASC labels. It would be helpful if the authors could relabel with more descriptive/widely used ASC population names such as those proposed in PMID 33148396 or PMID 36889280.

Response to Reviewers' comments. Reviewer comments are shown in full. Our responses in blue.

Reviewer #1 (Remarks to the Author):

This well written manuscript describes the transcriptomes at a single-cell level of CD45⁺ cells from the inguinal adipose depot of Pdgfrb-GFP transgenic mice. Analysis is complemented by comparison of iWAT fibroblast transcriptomes with existing datasets from heart and muscle, findings that iWAT fibroblasts are enriched in genes encoding for extracellular matrix. Variation due to sex, including Hox gene expression, are also observed. The authors confirm previously reported variation in in-vitro differentiation capacity related to cell surface expression of DPP4. In addition, the authors perform high quality immunofluorescence analysis of diverse progenitor populations in iWAT, at a single time point, finding varied localizations relative to the vasculature. In general, the results are very high quality, clearly presented, and consistent with the authors conclusions. However, there is little mechanistic or conceptual advancement over multiple already available studies of adipose progenitor cells in mice.

Reply: We thank the reviewer for the positive critique regarding data quality and presentation.

We would like to begin by pointing out that we had erroneously stated in the previous submission that the anatomical origin of cells for scRNA-seq and tissue images was inguinal WAT. The correct source was perigonadal WAT. This has now been corrected. It does not change any of the conclusions, except for the positive that certain markers show improved coherence between the paper's different datasets.

While we agree that multiple studies of adipose progenitor cells in mice have been published already, we believe that our present work confirms, balances and extends previous work by others thereby contributing to consensus as well as highlighting areas of conflict that require more work. Importantly, we provide to our knowledge the first comprehensive analysis focusing on sex difference as the key variable of ASC at the single-cell level. Our work thus highlights the importance of considering sex in future experimental studies of adipogenesis. The present submission is extensively revised, including new experiments and functional analyses. The conceptual advance that we now provide is that the sexual dimorphism observed in ASC in vivo is largely lost in primary cell culture. We have also added data on differences between subcutaneous and visceral WAT depots. We agree that molecular mechanistic insight from the transcriptomic information is still mainly at the level of "hypothesis generation", but to this end, the revised manuscript at least provides more analysis of context-dependent gene expression (by e.g. sex, depot, in vivo-ex vivo).

Specific comments:

1. The authors state that iWAT fibroblasts differ from muscle fibroblasts due to expression of adipocyte specific genes such as adiponectin and perilipin. These genes are indicative features of pre-adipocytes, such that cells expressing them should not be categorized as fibroblasts.

Reply: We understand the reviewer's point and suggest, as a compromise, to refer to ASC as fibroblast-like. The reasons are several: 1) they express the typical set of canonical markers of fibroblasts (*Pdgfra*, *Col1a1*, *Dcn*, *Lum* and others) which is a commonly accepted (albeit not perfect) characteristic of fibroblasts. 2) scRNA-seq data show that fibroblasts have a common set of characteristics that distinguish them from other major classes of cells, including their nearest relatives, mural and endothelial cells. All of these are strongly organotypic, although they retain generic hallmarks of the respective cell class. 3) Non-WAT fibroblasts, such as vascular adventitial fibroblasts (or fibroblast-like

cells) have mesenchymal stem cell characteristics and generate multiple differentiated connective tissue cell types, including adipocytes (see for example work from Bruno Peault and colleagues). 4) Besides ASC, we do not find any other fibroblasts in WAT. 5) The organotypic expression of matrisome genes by the ASC indeed suggest that they are the main producers and tailors of the WAT connective tissue—an archetypical function of fibroblasts. 6) By calling them fibroblast-like we wish to distinguish them from pericytes. Notably, WAT pericytes are typical pericytes in their gene expression and distinct from ASC.

In summary, while we agree about a preadipocyte role for ASC, they also seem to fulfill a role as tissue fibroblasts. Combined with the ubiquitous expression of canonical fibroblast markers, we propose a dual role: a stem cell pool with self-renewal properties and potential to differentiate into mature adipocytes, and organotypic WAT fibroblasts that tailor WAT extracellular matrix. We now use the term *fibroblast-like* for the ASC initially and conjunction with discussing stromal cell classes in WAT. We conclude that the gene signature of our fibroblast-like cell clusters fit the previous ASC classification (PMID:32026949), and thereafter use the ASC terminology for these cells in the subsequent sections of the paper.

2. Similarities between CD45- cells of adipose tissue and muscle could be attributed to the presence of adipocytes amongst muscle fibres.

Reply: We agree that adipocytes are often found interspersed between skeletal muscle fibers, but because there is already a lot published on the adipogenic potential of skeletal muscle fibroblast subpopulations (for example PMID: 37599828; PMID: 34210364; PMID:33529374), we have not elaborated specifically on the similarity between ASC and our previous skeletal muscle data in the manuscript. That said, our endomysial cluster 1 from (PMID:32769974) shares markers with ASC2, including *Pi16*, *Dpp4* and *Cd55*. This is worthy of study but, we feel, goes beyond the scope of our present manuscript.

Reviewer #2 (Remarks to the Author):

This work by Uhrbom et al uses single cell RNA sequencing to study adipose stem cell populations and examine the differences in populations across sex. Their major findings are:

- There are multiple populations of fibroblasts and mural cells in the adipose tissue.
- There is sexual dimorphism in gene expression and some functionality of adipose stem cells.
- Detailed imaging of adipose tissue shows that some ASC and pericyte populations are located near the adipose tissue vasculature.

Overall, the questions here are interesting, and the authors provide a valuable resource in their deeper sequencing of adipose stem cells using smart-seq. However, the paper is disjointed and almost feels like two studies (one on sexual dimorphism in ASCs and one on imaging of the adipose tissue vasculature) were merged into one paper. Additionally, a number of the findings they present (namely points 1 and 3 above) have already been made in other works, something the authors do not mention at all earlier in the work, presenting the findings as if they are novel and in one case describing previous literature as “scant”, but then the discussion section almost reads like a review article of previous work that their work now agrees with. This paper would be improved by focusing on the novelty in the paper (the characterization of the sexual dimorphism) with more functional assays. Specific comments are as follows:

Reply: We truly appreciate the reviewer’s constructive comments and feedback, which has helped us to structure the manuscript better and consolidate its message.

We would like to begin by pointing out that we had erroneously stated in the previous submission that the anatomical origin of cells for scRNA-seq and tissue images was inguinal WAT. The correct source was perigonadal WAT. This has now been corrected. It does not change any of the conclusions, except for the positive that certain markers show improved coherence between the paper's different datasets.

In retrospect, we understand the reviewer's concerns and are confident that we have now "joined" the parts better, as briefly outlined here:

The **first part** describes the overall landscape of non-parenchymal cells (i.e. all cells except the mature adipocytes) in WAT and proposes, based on single-cell transcriptomics, that adipose stem cells (ASC) are fibroblast-like cells (not pericytes) with a dual role as pre-adipocytes and extracellular matrix producing cells. We emphasize their organotypic gene expression profile and the distinction to mural cells. We agree that most of this information could have been extracted from previous publications on WAT scRNA-seq, but we know from own experience that different cell types may be variably represented in scRNA-seq data generated using different protocols and by different labs. Indeed, we note some variation in this regard. The first part sets the stage for the second and third parts by contributing to consensus, as well as highlight conflicting views (e.g. regarding Aregs), about ASC that require more work.

The **second part** concerns sexual dimorphic properties of ASC. Here, we confirm the sexually dimorphic gene profile using two different experimental protocols (scRNA-seq and bulk RNA-seq) and reanalyze publicly available datasets to further validate the findings. We explore some of the traits associated with sex-specific gene expression and assess their adipogenic capabilities in vitro, an ability that has previously been reported to differ between sexes in vivo.

The **third part** focuses on visualization of ASC subtypes in relationship to blood vessels. We acknowledge the reviewer's comment (below) that no sex differences were presented for this part in the previous submission, and we have therefore investigated this further and included new images displaying sex-specific expression at the protein level.

Regarding **novelty** of our findings, we apologize for presenting our WAT scRNA-seq data as if they were the first of its kind. The previous major WAT scRNA-seq papers are now mentioned and cited already in the Introduction, and a more detailed comparison is provided in the Discussion. We regret our previous use the word "scant" (now removed) for previous literature on the topic. Because discrimination between pericytes, vascular smooth muscle cells and fibroblast in tissue images is problematic (due to lack of cell specific markers), our tissue analysis using marker combinations visualizes morphological differences along the arteriovenous-axis for correct identification of cell types. The level of detail in these images brings our understanding of WAT perivascular cells to a new level, we think. Finally, our work has provided a more comprehensive analyses of sex differences in ASC-biology at the single cell level than previously reported. Replies to the specific comments can be found below.

1. Comment: The pathway analysis in figure 3 suggests different pathways may be active in male vs female ASCs. Some functional characterization of these pathways in ASCs from both sexes would enhance the paper, such as seahorse or clark electrode measurement of OxPhos.

Reply: Thank you, this is a valid comment. To investigate if the difference in mitochondrial gene expression also resulted in a functional difference, we assessed the mitochondrial capacity in vitro using the Seahorse XFe96 analyzer on confluent crude SVF cells isolated from perigonadal (pg) and inguinal white adipose tissue (iWAT) of both male and female mice. The results **did not** indicate a difference in the rate of ATP production or maximal respiratory capacity in crude stromal vascular fraction (SVF) cells between sexes (see Fig. 1 below). While this initially boggled our minds, we learned using bulk RNA-seq analysis of confluent SVF cells cultured for 4 days (i.e. under identical condition as cells used for the

functional analysis) that the sexually dimorphic expression of mitochondrial and nuclear encoded ETC-genes was lost in culture (see Fig. 2 below) likely explaining the negative Seahorse data. Also the sexually dimorphic core set of 19 genes was lost, which (see new Supplementary Figs. 6c-g, described on line 259-279 in the text). We used crude SVF cells for these assays because it turned out unfeasible to collect sufficient ASC by FACS for this type of analysis. FACS is also harsh on the cells. Why the sexually dimorphic gene expression patterns are lost *in vitro* is unclear, but may reflect multiple changes in the microenvironment, including soluble and cell- and matrix associated signals and changes in cell shape. That said, the high expression levels of fibroblasts markers (various collagens, *Dcn*, *Lum*, *Pdgfra*, *Fn1*, and others) and low expression of markers for endothelial cells, macrophages, pericytes or VSMC in the SVF culture indicated that they were dominated by ASC (Supplementary Fig. 6j). We tried to perform mitochondrial capacity measurements 24h after the cells were seeded to increase the chance of detecting functional difference in mitochondrial activity, but the cells did not respond to the reagents (Oligomycin, FCCP and Rotenone/Antimycin) sufficiently for meaningful data, perhaps because of the stress experience from detachment from the *in vivo* context.

In **brief summary**, we tried, but failed, to assess functional consequences of the sexually dimorphic ASC phenotype *in vitro*, but there is a likely reason for the failure: the sexually dimorphic Oxphos-gene expression gets lost in ASC *in vitro*. We show this by gene expression data in the revised manuscript, but we do not show the negative functional data, as they merely confirm what could be expected.

Mitochondrial stress test on SVF cells from pgWAT and iWAT

Figure 1, Seahorse Mitochondrial stress test on *in vitro* cultured SVF from pgWAT and iWAT, error bars represent standard deviation

Reads from ETC-genes from cultured crude SVF cells

Figure 2, Percentage of read counts of total reads from mitochondrial and nuclear encoded electron transport chain (ETC)-genes

As for the male-enriched RAAS system we conducted experiments investigating the effect of Angiotensin II (AngII), which has been reported to impact both lipolysis and the differentiation of ASC. However, our experiments using 100 ng/ml of AngII did not show any effect on basal lipolysis in iWAT explants (Fig. 3h) or differentiation capacity of SVF cells from the same depot (Fig. 3I). Of note, transcripts from *Agtr1a* and *Agtr2* genes were detected in confluent crude SVF cells at start of differentiation (Fig. 3J). This is now described in the manuscript (line 218-224). We used iWAT explant because all cell-types of the adipose depot are present providing a potential to capture cellular cross-talk impacting lipolysis after AngII-stimuli. Even this was long shot, however, since vascular cells, although present in the explants, are uncoupled from the circulation.

2. Comment: When the authors differentiate male and female preadipocytes in culture, do they observe sexually dimorphic gene expression in the mature adipocytes? If so, how similar is it to what they see in the preadipocytes?

Reply: This is a good question, but a challenging one to address experimentally. As discussed above, the sexually dimorphic gene expression signature was lost with time in culture for isolated SVF cells, this is most likely true also for isolated FACS sorted DPP4- (ASC1a/b) cells, since the dominant cell-type in SVF cultures are the ASC. The question is if the ASC retain their sexually dimorphic gene expression signature following adipogenic differentiation. The simple answer is most likely “NO” given the argument above.

3. Comment: It is interesting that the authors see a sexually dimorphic difference in cultured ASCs from male and female animals, and raises the question of what differences are preserved in culture and what are not. If ASCs from male and female animals are cultured for say a week, do they still see the transcriptional differences that they see in freshly isolated ASCs?

Reply: As discussed above the sexually dimorphic 19-gene signature gets lost in crude SVF cells already after four days in culture (Supplementary fig 6c-g), which is now discussed in the revised manuscript (line 259-279).

4. Comment: The cells in clusters 9 and 13 were correctly removed by the authors in the analysis done in figures 1 and 2 because of high mitochondrial gene expression, which is indicative of low-quality cells. Why then did they re-include these cells for the analysis in figure 3? These cells should be removed from all analyses.

Reply: Clusters 9 and 13 were both removed from the analysis and not included in Figure 3 as pointed out in the text line 92-94 and 205-205. "Of note, the mitochondrial gene-enriched EC and ASC cluster 9 and 13, respectively, had been excluded from this analysis " .

5. Comment: The authors describe the current literature on adipose tissue microvasculature as "scant", yet there have actually been a number of papers published on the subject, including one also showing that Pdgfrb+ cells are located near the vasculature (PMID 26626462) and one that shows that DPP4+ ASCs are also located near the vasculature (PMID 36168895). The authors should significantly modify their discussion of this data to include some of the large body of work that has been done to study the adipose tissue vasculature.

Reply: We thank the reviewer for this comment. We have made extensive textual changes to present our contribution to the field, building on and extending existing data and concepts. The rapid advance in scRNAseq technology and application has certainly transformed our understanding of the cellular composition of adipose tissue, which might be one reason why we have arrived at different interpretations than some of the previous investigators.

6. Comment: The inclusion of figures 5 and 6 in this paper are confusing, as they are not related to the sexual dimorphism of adipose precursor cells that the rest of the paper is focused on. It is especially confusing to see a large diagram in figure 6c that is only related to the last two figures of the paper and not a summary of the entire paper. Is there sexual dimorphism in any of the findings shown here?

Reply: We have revised the text extensively to better integrate the tissue morphology analysis shown in Figs 5 and 6 into the manuscript narrative. We also provide a new piece of data to this end: evidence for sexually dimorphic expression of nerve growth factor receptor (NGFR, encoded by the female-specific transcript *Ngfr*) which was stained by immunofluorescence in ASC in pgWAT only in females in line with the mRNA data (Figs. 6c-d). Other tested antibodies failed to provide reliable patterns of immunofluorescent staining. *Ngfr* is one of the 44 differentially expressed genes for pgWAT (Fig. 2c). These results are now included in the paper and new text describing this can be found on line 308-314.

7. Comment: The authors describe in detail removing pericytes with suspected endothelial cell transcript contamination, but this could be the result of ambient RNA, a package designed to computationally remove ambient RNA like cellbender should be used on the data to see if this allows them to remove potentially contaminating transcripts from their dataset.

Reply: We are skeptical to computational filtering, because we think it is difficult to classify ambient RNA. Regardless, pericyte-endothelial cross contamination is notorious and in our experience based on cell fragment contamination rather than ambient RNA contamination, which allows identification and

discrimination between contaminated versus non-contaminated cells. See our previous publications (PMID: 33405941; PMID: 35452595) where we discuss this in detail.

8. Comment: The utility of most of the data in figure 1 (the dotplot in panel c, panels d&f) is a little unclear. There has been extensive work already on characterizing adipose fibroblasts at single cell resolution (for example, see PMID 33981032 for comparison between adipose and other fibroblasts, see PMID 36889280 for a review of adipose fibroblasts), and so it's a little confusing to find things like enriched genes and pathways in adipose fibroblasts compared to other fibroblasts presented in a main figure like this. I appreciate that the authors want to pull in data from their previous work and don't have a problem with the analysis itself, but the way this data is presented ignores a lot of previous work on the subject. The discussion does have almost a review of some of the previous work but the authors should also modify the results section to better place their work in the context of previous work.

Reply: As mentioned above, we agree with this comment and have thoroughly acknowledged and discussed the previously published scRNA-seq data on WAT. Our idea is now to first classify the ASC in a wider context and show that they are similar to fibroblasts in other organs although they possess an organotypic gene expression profile (Figs. 1d-e, Supplementary Figs. 1b-d). A change that we have done, which brings validation to our observations and puts our analysis in the context of previous published work was to include bar plots of the expression of the enriched genes in adipose fibroblasts based on the data from PMID 33981032. As seen in the Supplementary Fig. 1d, almost all the enriched adipose fibroblast genes associated with lipid metabolism in our analysis are also enriched in adipose fibroblasts in PMID 33981032. Text describing this can be found on line 111-112.

9. Comment: Currently the labeling of ASCs varies widely by study, but there has been some effort to create consensus ASC labels. It would be helpful if the authors could relabel with more descriptive/widely used ASC population names such as those proposed in PMID 33148396 or PMID 36889280.

Reply: We use the nomenclature proposed by Elisabeth A. Rondini and James G. Granneman (PMID: 32026949) throughout and now state this clearly. Some of the other labels includes functional descriptions, such as Aregs (Adipogenesis regulating cells, named ASC1b in our paper), which in our view is suboptimal since consensus has not been reached about the function of the different ASC subtypes.

REVIEWER COMMENTS

Reviewer #1 (Remarks to the Author):

The authors have greatly improved the quality of the manuscript with extensive re-writing and additional experiments. I have no further concerns.

Reviewer #2 (Remarks to the Author):

I appreciate the authors engagement with my comments and the revisions they have made to the manuscript. I do still have a couple of concerns, as follows:

1. Like I said, I appreciate the authors responses to my comments, including the new analysis in supplementary figure 6 showing loss of sex-specific gene expression in cultured ASCs, making it more difficult to do functional analysis of these differences. I do wonder though if it would still be possible to measure oxygen consumption in freshly isolated ASCs in order to test for a sex-specific difference before they dissipate in culture.
2. I apologize for not mentioning this in the previous review, but the DEA analysis the authors have done using the Wilcoxon rank-sum test is not best practice, and results in artificially inflated p values like those seen in the volcano plots in Figures 3b and 4d. I suggest using a pseudobulk method for this analysis, please see PMID 34584091.
3. The labeling of male and female ASC subclusters in figure 1 (c,f, and g) is a little confusing to me, I would suggest labeling more like that in figure 4g.

REVIEWER COMMENTS

Reviewer #1 (Remarks to the Author):

The authors have greatly improved the quality of the manuscript with extensive re-writing and additional experiments. I have no further concerns.

We thank reviewer #1 for the help to improve our study.

We would like to point out that minor adjustments have been done in the manuscript apart from the changes described in the replies to Reviewer #2's questions. These modifications can be found on line 305-309 (see new figure 5e and 6c) and concerns the mesothelial layer of pgWAT.

Reviewer #2 (Remarks to the Author):

I appreciate the authors engagement with my comments and the revisions they have made to the manuscript. I do still have a couple of concerns, as follows:

Apart form the replies below, we would like to point out some additional minor adjustments in the manuscript. These modifications can be found on line 305-309 (see new figure 5e and 6c) and concerns the mesothelial layer of pgWAT.

1. Like I said, I appreciate the authors responses to my comments, including the new analysis in supplementary figure 6 showing loss of sex-specific gene expression in cultured ASCs, making it more difficult to do functional analysis of these differences. I do wonder though if it would still be possible to measure oxygen consumption in freshly isolated ASCs in order to test for a sex-specific difference before they dissipate in culture.

We performed the suggested experiment, namely to measure oxygen consumption levels in freshly isolated ASC (CD45-/CD31-/TER119-/CD146- fraction of SVF) from pgWAT. The experiment (see Figure 1 for reviewers below) provided a clear result but did not show any difference in the maximal respiration or ATP production between male and female ASC. The measurements were made on a Seahorse instrument within 24 hours of isolation and after cells have attached to the bottom of the plate. Setting up a functional protocol for freshly isolated cells in suspension would obviously be desirable, but the time and efforts required are unpredictable and therefore go beyond the scope and timeline for the final revisions of this manuscript.

Freshly isolated ASC from pgWAT

CD45-/CD31-/TER-119-/CD146- fraction of SVF

Figure 1. for reviewer: Seahorse measurement of Oxygen consumption levels using the Mitochondrial stress test assay on freshly isolated ASC cells from pgWAT. The assay was conducted on cells within 24 hours of isolation. The ASC were isolated by depleting the CD45/CD31/TER119/CD146 fraction from the stromal vascular fraction of pgWAT.

2. I apologize for not mentioning this in the previous review, but the DEA analysis the authors have done using the Wilcoxon rank-sum test is not best practice, and results in artificially inflated p values like those seen in the volcano plots in Figures 3b and 4d. I suggest using a pseudobulk method for this analysis, please see PMID 34584091.

As suggested, we performed DE-analysis on our scRNA-seq dataset using the pseudo bulk EdgeR-LRT method. This gave similar results as with the Wilcoxon rank-sum test (see Figure 2 for reviewers below and the updated Fig 2 in the revised manuscript), but the results highlighted two things.

Firstly, the DE-genes resulting from the comparison between the two bulk RNA-seq dataset from pgWAT and iWAT (red-colored genes in Figure 2 c,d) provide an extended list of possible sexually dimorphic DE-genes. We have now described this with new text on line 158-160 in the manuscript and we have modified Figure 2c in the manuscript accordingly.

Secondly, only a minority of the OxPhos-genes were statistically significant different in the pseudo bulk EdgeR-LRT analysis. Thus the broader difference in OxPhos-gene expression is only observed in our scRNA-seq dataset. Other observations like the *Hox* gene expression pattern, the core set of 19 sexually dimorphic genes and the majority of the 44 pgWAT specific DE-genes were all statistically significant DE-genes in the EdgeR analysis (Figure 2, c,d). We have made some adjustments to the text, mentioning that the broader differences in OxPhos gene expression is observable only in our scRNA-seq dataset (see line 209-212). Moreover, the difference Oxphos-gene expression is no longer mentioned in the abstract.

c DE-analysis based on Wilcoxon Rank-sum test using Bonferroni adjusted p-values

d Pseudo bulk DE analysis with EdgeR-LRT using Benjamini-Hochberg adjusted p-values

Figure 2 for reviewer: Comparison of Wilcoxon rank-sum test versus Pseudo bulk DE-analysis using EdgeR-LRT, a. Volcano-plot of DE-genes between male and female ASC based on Wilcoxon rank sum-test. B. same as in a but for pseudo bulk DE-analysis with

EdgeR-LRT c. Original Venn diagram for which the DEGs in the scRNA-seq data are obtained using the Wilcoxon rank-sum test. d. same as in c but DEGs are derived from pseudo bulk DE analysis with EdgeR-LRT. Genes colored in red shows DE-genes restricted to the bulk RNA-seq samples in the original Venn diagram.

3. The labeling of male and female ASC subclusters in figure 1 (c,f, and g) is a little confusing to me, I would suggest labeling more like that in figure 4g.

We agree and have changed Figure 1 according to the reviewer's suggestion.

REVIEWERS' COMMENTS

Reviewer #2 (Remarks to the Author):

I again appreciate the authors willingness to engage with my comments. I am glad they redid the DEA using pseudobulk and I think that these should be the results they present in figures 2b and 3d. In their response, the authors state "only a minority of the OxPhos-genes were statistically significant different in the pseudo bulk EdgeR-LRT analysis. Thus the broader difference in OxPhos-gene expression is

only observed in our scRNA-seq dataset" but this is not entirely accurate, the pseudobulk analysis IS the results from the scRNA-seq dataset, it is a more robust analysis that's less prone to false discovery. The way the authors currently present the data is as if they did two separate experiments that show some differences in results, when in fact they applied two methods to one dataset and are choosing to preferentially display the less accurate one. I appreciate that the Wilcoxon based DEA shows a difference in OxPhos whereas the pseudobulk DEA does not and that the authors want to preserve that finding. While I know there are many technical limitations to studying the difference in OxPhos, the fact that the authors have been unable to see an experimental difference and the fact that those genes are not all showing up in the new analysis might suggest that OxPhos might not actually be different here. I appreciate that the authors have already softened their language on this point and I am not insisting that they take any mentions of OxPhos differences out entirely, but I do think it is important that they also show the more accurate plots and analysis in their figures.

Reviewer #2 (Remarks to the Author):

I again appreciate the authors willingness to engage with my comments. I am glad they redid the DEA using pseudobulk and I think that these should be the results they present in figures 2b and 3d. In their response, the authors state "only a minority of the OxPhos-genes were statistically significant different in the pseudo bulk EdgeR-LRT analysis. Thus the broader difference in OxPhos-gene expression is only observed in our scRNA-seq dataset" but this is not entirely accurate, the pseudobulk analysis IS the results from the scRNA-seq dataset, it is a more robust analysis that's less prone to false discovery. The way the authors currently present the data is as if they did two separate experiments that show some differences in results, when in fact they applied two methods to one dataset and are choosing to preferentially display the less accurate one. I appreciate that the Wilcoxon based DEA shows a difference in OxPhos whereas the pseudobulk DEA does not and that the authors want to preserve that finding. While I know there are many technical limitations to studying the difference in OxPhos, the fact that the authors have been unable to see an experimental difference and the fact that those genes are not all showing up in the new analysis might suggest that OxPhos might not actually be different here. I appreciate that the authors have already softened their language on this point and I am not insisting that they take any mentions of OxPhos differences out entirely, but I do think it is important that they also show the more accurate plots and analysis in their figures.

Comment:

We appreciate the further comment by Reviewer #2, and we agree. Because we have no other data to support a difference in the expression of Oxphos related genes between the sexes than the Wilcoxon based DE-analysis, and, as the reviewer points out, because this is less reliable than the pseudo-bulk analysis, we decided to present the pseudo bulk DE-analysis only and to remove the Oxphos-related data and its discussion from the manuscript. The resulting changes to the manuscript can be seen on line: 68, 154-164, 171-173,204-220, 276, 256-360 and in Figure 2, 3 and in Supplementary Information file.